

# Evaluation of radiation schemes in the CMA-MESO model using high time-resolution radiation measurements in China: I. Long-wave radiation

Junli Yang[1,2,6], Weijun Quan[3,4,5], Li Zhang[1,2,6], Jianglin Hu[1,2,6], Qiying Chen[1,2,6], and Martin Wild[4]

[1]CMA Earth System Modelling and Prediction Centre (CEMC), China Meteorological Administration, Beijing, 100081, China

[2]Key Laboratory of Earth System Modelling and Prediction, China Meteorological Administration, Beijing, 100081, China

[3]Institute of Urban Meteorology (Key Laboratory of Urban Meteorology), China Meteorological Administration, Beijing, 100089, China

[4]Institute for Atmospheric and Climate Science, ETH Zürich, Zürich, 8092, Switzerland

[5]Beijing Weather Forecast Centre, Beijing Meteorological Service, Beijing, 100097, China

[6]State Key Laboratory of Severe Weather (LaSW), China Meteorological Administration, Beijing, 100081, China

*Correspondence to*: Weijun Quan (quanquan78430@163.com) and Martin Wild (martin.wild@env.ethz.ch)

**Abstract.** Downward long-wave irradiance (DnLWI) is a variable that directly influences the surface net radiation, which in turn affects weather and climate. Due to the lack of abundant DnLWI observations, numerical weather prediction (NWP) models provide a very effective way to yield the DnLWI. Nevertheless, the reliability of the DnLWI predicted by the NWP models needs the evaluation based on the observations or accurate radiative transfer models. In this study, the DnLWI product of the China Meteorological Administration mesoscale model (CMA-MESO) was extensively validated using long-term high time-resolution (1 min) DnLWI measurements carried out at 42 sites in China. The results showed that the DnLWIs predicted by the CMA-MESO model generally agreed well with the observations, i.e., with a relative mean bias error (rMBE) of –2.0%, but overestimated them under overcast (3.1%) and underestimated them under dry (–5.3%) and cloudless (–5.2%) conditions. It is also found that the discrepancies in the DnLWI predicted by the CMA-MESO model exhibited spatial and diurnal variations, e.g., the discrepancies were significantly smaller at night than those during the day due to the stable nocturnal boundary layer. According to the results of the partial least squares analysis, the high cloud cover, medium cloud cover, planetary boundary layer height, and integrated cloud ice were the most important factors affecting the accuracy of the DnLWIs predicted by the CMA-MESO model under all sky conditions. By comparing the outputs of the CMA-MESO model and the MODerate resolution atmospheric TRANsmission (MODTRAN) model, it was found that the uncertainties in the DnLWI predicted by the CMA-MESO model mainly arose from an inappropriate consideration of the high and medium clouds under all sky conditions as well as shortcomings in the radiation scheme under cold dry cloudless conditions.





## 1 Introduction

Downward long-wave irradiance (DnLWI) refers to the long-wave radiation in the range of 4.0–100.0 μm, which is mainly emitted by water vapour ($H_2O$), carbon dioxide ($CO_2$), and ozone ($O_3$) molecules as well as cloud water droplets in the sky
(Idso and Jackson, 1969). The DnLWI reaching the ground plays a substantial role in climate studies (Wild and Cechet, 2002), weather forecasting (Morcrette, 2002a), Earth-atmosphere energy exchanges (Idso and Jackson, 1969; Heitor et al., 1991) and even in energy-saving applications (Li et al., 2017). Because of the difficulties involved in the measurements of the DnLWI, this quantity has for long only been observed at a few sites (Wild et al., 2001; Duarte et al., 2006; Li et al., 2017; Yang et al., 2023). The main reason lies in the fact that the pyrgeometer is usually expensive and sensitive due to the
radiation emitted by the instrument body is comparable to that being measured in wavelength (Albrecht and Cox, 1977). To overcome this shortcoming, some proxies have been proposed to obtain the DnLWI indirectly, e.g., through parameterizations in terms of more readily available meteorological variables (Brunt et al., 1932; Duarte et al., 2006; Yang et al., 2023), simulations from radiative transfer models (Feigelson et al., 1991; Viúdez-Mora et al., 2009), predictions of numerical forecasting models (Morcrette, 1991), etc.

The reliability of the DnLWI predicted by numerical forecasting models depends largely on the adopted radiation transfer schemes as well as the vertical temperature and humidity profiles used as input to the radiation code. Up to now, numerous radiative transfer schemes have been developed for predicting the DnLWI. For instance, the radiative transfer scheme used in the European Centre for Medium Range Weather Forecast (ECMWF) operational model (Morcrette, 1991), the long-wave radiation scheme adopted by the Geophysical Fluid Dynamics Laboratory (Schwarzkopf and Fels, 1991), the
radiation scheme in the Community Climate System Model Version 3 (Collins et al., 2006), as well as the rapid radiative transfer model (RRTM, Mlawer et al., 1997) employed in the ECMWF model (Morcrette, 2002a). Note that the RRTM is also applied in the Weather Research and Forecast model (Iacono et al., 2008), as well as in the China Meteorological Administration mesoscale model (CMA-MESO) model, which is one of the successors of the Global/Regional Assimilation and PrEdictions System (GRAPES) independently developed by the CMA since the 2000s (Chen et al., 2008; Xu et al., 2008;
Xue et al., 2008; Huang et al., 2017; 2022; Ma et al., 2022; Shen et al., 2023). The RRTM scheme uses a correlated-$k$ method and look-up tables to accurately compute the long-wave fluxes and cooling rates over the long-wave spectral region (10–3000 $cm^{-1}$ or 3.333–1000 μm), which result from the absorption and emission of infrared radiation of $H_2O$, $O_3$, $CO_2$, methane ($CH_4$), nitrous oxide ($NO_x$), halocarbons (HCFs), and clouds (Shen et al., 2004). To run the RRTM in the CMA-MESO, some parameters (e.g., the air pressure, temperature, water vapour, mixing ratio of cloud water/ice, cloud cover, the
concentrations of $O_3$ and $CO_2$, and the surface emissivity) are required to input, but the effects of aerosols, $N_2O$, $CH_4$, CFC11, and CFC12 are not taken into account. Though developers have made a lot of efforts to improve the RRTM scheme in the CMA-MESO model, e.g., introducing the effects of geographical slopes into the radiation scheme (Xu et al., 2008), a systematic evaluation of radiation scheme is seldom conducted.





Evaluation and diagnosis of radiation schemes play a critical role in steadily optimizing and improving the Numerical
Weather Prediction (NWP) forecast systems. To this end, three methods are generally applied. The first method is to
compare the DnLWIs predicted by the radiation schemes with those simulated by a precise line-by-line radiative transfer
model (LBLRTM; Clough et al., 1992; Clough and Iacono, 1995). The LBLRTM is the most accurate radiative transfer
model for calculating the DnLWI, but it is limited by the precision of the input atmospheric profiles and its complexity for
practical applications (Li et al., 2017). The second method is to validate the DnLWI prediction from the radiation schemes
based on the corresponding outputs of the narrow-band model such as the MODerate resolution atmospheric TRANsmission
model (MODTRAN; Berk et al., 1999). The MODTRAN introduces a correlated-$k$ algorithm which significantly improves
the accuracy and speed of radiance calculations, particularly for multiple scattering in spectral regions containing strong
molecular line absorption (Berk et al., 1998). The reliability of the MODTRAN model essentially depends on the absorption
coefficients of atmospheric constituents and the atmospheric input profiles (Schweizer and Gautier, 1995). The third method
is to compare the DnLWI predicted by the radiation scheme with in-situ DnLWI observations of pyrgeometer (Wild et al.,
2001; Garcá et al., 2018). In spite of an independent information source provided by the pyrgeometer, the in-situ DnLWI
measurements are still limited due to their sparse spatial distribution and low temporal resolution (Wild et al., 2017).
Fortunately, A long-term high time-resolution (1 min) observational dataset of the DnLWI has been established based on
several observation networks in China, i.e., the global/regional Global Atmosphere Watch (GAW) stations (Song et al., 2013;
Quan et al., 2024), the China Baseline Surface Radiation Network (CBSRN) stations (Yang et al., 2023), the National
Climate Observatory (NCO) stations (Xu et al., 2013), and the Basic Meteorological Station (BMS). Thanks to their high
temporal resolution (1 min), the in-situ observations of the DnLWI allow to precisely match the hourly instantaneous
predictions of the DnLWI from the CMA-MESO model, which provides an opportunity to evaluate the forecast ability of the
radiation scheme in the CMA-MESO model over various underlying surfaces and climate zones.

In this study, the in-situ measurements of the DnLWI are employed to evaluate the results forecasted by the operational
CMA-MESO model. In particular, the potential influential mechanisms of the DnLWI prediction are also investigated via
comparing with the simulations of the MODTRAN model. This paper is organized as follows. Section 2 describes the
observational data, the model, and analysis methods. The evaluation results are shown in Section 3. The summary and
concluding remarks are given in the final section.

## 2 Observation, model, and methods

### 2.1 In-situ observation

In this study, high time-resolution (1 min) observations of the DnLWI carried out at 42 validation sites in China were used as
a reference dataset to evaluate the performance of the CMA-MESO model in predicting the DnLWIs. It can be seen from
Table 1 that the validation sites consist of three regional GAW stations (Longfengshan, Shangdianzi, and Lin'an), six
CBSRN stations (Yanqi, Xilinhot, Mohe, Xuchang, Wenjiang, and Dali), five NCO stations (Shenzhen, Sanxia, Nanchang,





Beihai, and Sanya), and twenty eight BMS stations (Golmud, Xining, Hulunbeir, Yuzhong, Altay, Tacheng, Urumchi, Kashi, Hetian, Dunhuang, Yinchuan, Taiyuan, Beijing, Heihe, Harbin, Shenyang, Jinan, Jinghe, Kunming, Wuhan, Shapingba, Changsha, Guiyang, Nanjing, Baoshan, Yong'an, Guangzhou, and Nanning). Note that, some stations belong to two networks simultaneously, e.g., the Shangdianzi station is one of regional GAW stations as well as a CBSRN station in China.


**Table 1.** Basic descriptions for the validation sites used in this study**.**

| Station name | Abbr. | Station ID | Latitude (° N) | Longitude (° E) | Altitude (m a.s.l) | Data period | Climatic zone |
|---|---|---|---|---|---|---|---|
| Golmud | GLM[*] | 52818 | 36.42 | 94.91 | 2807.6 | 202104–202212 | PLTC |
| Xining | XN[*] | 52866 | 36.66 | 101.73 | 2434.2 | 202104–202212 | |
| Yanqi | YQ[¤] | 51567 | 42.05 | 86.61 | 1056.5 | 201906–202212 | TPCC |
| Xilinhot | XL[¤※] | 54102 | 44.13 | 116.33 | 1107.0 | 201607–202108 | |
| Hulunbeir | HL[*] | 50527 | 49.25 | 119.70 | 649.6 | 202104–202212 | |
| Yuzhong | YZ[*] | 52983 | 35.87 | 104.15 | 1874.4 | 202105–202209 | |
| Altay | ALT[*] | 51076 | 47.73 | 88.08 | 735.3 | 202105–202212 | |
| Tacheng | TC[*] | 51133 | 46.73 | 83.0 | 534.9 | 202104–202212 | |
| Urumchi | URM[*] | 51463 | 43.78 | 87.65 | 935.0 | 202104–202212 | |
| Kashi | KS[*] | 51709 | 39.49 | 75.75 | 1384.4 | 202104–202212 | |
| Hetian | HT[*] | 51828 | 37.13 | 79.93 | 1375.0 | 202104–202212 | |
| Dunhuang | DH[*] | 52418 | 40.15 | 94.68 | 1139.0 | 202104–202212 | |
| Yinchuan | YC[*] | 53614 | 38.47 | 106.21 | 1111.6 | 202104–202212 | |
| Taiyuan | TY[*] | 53772 | 37.62 | 112.58 | 776.3 | 202104–202212 | |
| Longfengshan | LFS[†] | 54084 | 44.73 | 127.60 | 331.0 | 201910–202212 | TPMC |
| Shangdianzi | SDZ[†¤] | 54421 | 40.65 | 117.12 | 293.3 | 201607–202212 | |
| Mohe | MH[¤] | 50136 | 52.97 | 122.52 | 438.5 | 202104–202212 | |
| Beijing | BJ[*] | 54511 | 39.48 | 116.28 | 31.3 | 202104–202212 | |
| Heihe | HH[*] | 50468 | 50.25 | 127.46 | 166.4 | 202104–202212 | |
| Harbin | HEB[*] | 50953 | 45.93 | 126.57 | 118.3 | 202105–202212 | |
| Shenyang | SY[*] | 54342 | 41.73 | 123.51 | 49.0 | 202105–202212 | |
| Jinan | JN[*] | 54823 | 36.6 | 117.01 | 170.3 | 201905–202212 | |
| Jinghe | JH[*] | 57131 | 34.45 | 108.97 | 407.6 | 202104–202212 | |
| Lin'an | LAN[†] | 58448 | 30.28 | 119.75 | 138.6 | 201607–201912 | STMC |
| Xuchang | XC[¤] | 57089 | 34.07 | 113.93 | 67.2 | 201607–202202 | |
| Wenjiang | WJ[¤※] | 56187 | 30.75 | 103.86 | 547.7 | 201607–202212 | |
| Dali | DL[¤※] | 56751 | 25.71 | 100.18 | 1990.5 | 201906–202108 | |
| Shenzhen | SZ[※] | 59486 | 22.65 | 113.89 | 50.1 | 201607–202108 | |



| Kunming | KM[*] | 56778 | 25.01 | 102.65 | 1888.1 | 202101–202212 | |
| Sanxia | SX[※] | 57461 | 30.74 | 111.36 | 256.5 | 202101–202212 | |
| Wuhan | WH[*] | 57494 | 30.6 | 114.05 | 23.6 | 201903–202212 | |
| Shapingba | SPB[*] | 57516 | 29.58 | 106.47 | 259.1 | 202104–202212 | |
| Changsha | CS[*] | 57687 | 28.11 | 112.79 | 119.0 | 202104–202212 | |
| Guiyang | GY[*] | 57816 | 26.59 | 106.73 | 1227.3 | 202104–202212 | |
| Nanjing | NJ[*] | 58238 | 31.93 | 118.9 | 35.2 | 202106–202212 | |
| Baoshan | BS[*] | 58362 | 31.39 | 121.44 | 3.3 | 202002–202212 | |
| Nanchang | NC[※] | 58606 | 28.59 | 115.9 | 47.2 | 202104–202212 | |
| Yong'an | YON[*] | 58921 | 25.97 | 117.35 | 206.0 | 202107–202212 | |
| Guangzhou | GZ[*] | 59287 | 23.21 | 113.48 | 70.7 | 202104–202212 | |
| Nanning | NN[*] | 59431 | 22.78 | 108.55 | 152.1 | 202104–202212 | |
| Beihai | BH[※] | 59644 | 21.45 | 109.18 | 10.3 | 202101–202108 | TRMC |
| Sanya | SAY[※] | 59948 | 18.22 | 109.59 | 419.4 | 202104–202212 | |

[†]Global Atmosphere Watch station; [☆]China Baseline Surface Radiation Network station; [※]National Climate Observatory; [*]Basic Meteorological Station.

China is a large country, in which about 98% of the land area stretches between latitudes of 20°N to 50°N, from subtropical zones in the south to the temperate zones. In addition, China is situated between Eurasia and the Pacific Ocean,

allowing the development and prevalence of monsoon, i.e., with a marked change in wind direction between winter and summer as well as seasonal variations in precipitation (Lam et al., 2005). China also has a complex topography ranging from mountainous regions to flat plains, which combined with the unique geographical location of China to come into being several different climate zones. In this study, five climatic zones are categorized, i.e., the temperate continental climate (TPCC), the temperate monsoon climate (TPMC), subtropical monsoon climate (STMC), the tropical monsoon climate

(TRMC), and the plateau climate (PLTC). The TPCC is characterized by cold winter, hot summer, and large annual temperature differences as well as small precipitation. Hot and rainy in summer, cold and dry in winter are the major climatic characteristics of the TPMC. The STMC is hot in summer and warm in winter, with four distinct seasons and developed monsoon. The TRMC is characterized by high temperature and high rainfall throughout the year. As the temperature decreases with the increase of height, the high altitude in the PLTC arises a low temperature throughout the year.

Several types of pyrgeometer (see Table 2) were adopted at the validation sites, which were carefully maintained to provide precise 1-min resolution data of the DnLWI, e.g., the CG4 pyrgeometer (Kipp & Zonen, the Netherlands), the IR02 pyrgeometer (Hukseflux, the Netherlands), the PIR pyrgeometer (the Eppley Laboratory, U.S.A.), the FS-T1 pyrgeometer (Jiangsu Radio Science Institute, China), and the HTL-2 pyrgeometer (Huatron Co. Ltd., China). The spectral ranges of these instruments cover most of the spectrum of long-wave radiation emitted by the atmosphere and ensure them to be suitable

instruments for measuring the DnLWI. Note that the field of view (FOV) of the IR02 is 150° rather than the desired 180°,





which causes its price to be attractive while the accuracy loss is relatively minor (Hukseflux, 2022). To eliminate the influence of solar radiation, the pyrgeometer is shaded by a ball mounted on the automatic solar tracker during observation (Yang et al., 2023). In addition, a ventilation-heating system is installed to reduce the influence of environmental temperature and to prevent dew/dust fall on the dome of the pyrgeometer. The pyrgeometer samples at a frequency of 1 Hz, and the 1 min
averaged data are stored using a data logger. To assure the DnLWI measurements from the pyrgeometer can be traceable to the World Radiometric Reference like those observed at the Baseline Surface Radiation Network (Driemel et al., 2018), the pyrgeometers used in this study were calibrated against the reference CGR 4 pyrgeometer of the CMA, which is traced to the World Infrared Standard by participating in the International Pyrgeometer Comparison organized by the Physikalisch-Meteorologisches Observatorium Davos and the World Radiation Center (Gröbner et al., 2014; Gröbner and Thomann, 2023;
Yang et al., 2023).

**Table 2.** Specifications of the pyrgeometer used in this study.

| Instrument Type | Spectral range (μm) | Response time (s) | Non-linearity (% yr$^{-1}$) | Field of View (°) | Manufacture |
|---|---|---|---|---|---|
| CG4 | 4.5–42.0 | < 18 | < ±1 | 180 | Kipp & Zonen, the Netherlands |
| IR02 | 4.5–42.0 | < 18 | < ±2.5 | 150 | Hukseflux, the Netherlands |
| PIR | 4.0–50.0 | < 5 | < ±0.5 | 180 | Eppley Laboratory, U.S.A. |
| FS-T1 | 4.5–50.0 | < 20 | < ±2 | 180 | Jiangsu Radio Science Institute, China |
| HTL-2 | 4.0–50.0 | < 30 | < ±2 | 180 | Huatron Co. Ltd., China |

The 1-min raw radiometric data used in this study underwent strict quality control tests by using the Hybrid Algorithm for Radiation Data Quality Control (HARDQC), in which the downward long-wave irradiance is designed to pass three tests, i.e., the "Physically possible limits test", the "Extremely rare limits test", and the "Comparison test between radiation
components" (Quan et al., 2024). The 1-min downward long-wave irradiance data, which have passed all three tests, were selected as the references to validate the DnLWIs predicted by the CMA-MESO model.

## 2.2 CMA-MESO model

The CMA-MESO model has undergone seven upgrades since 2014, i.e., Version 4.0 (release date: 10 July 2014), Version 4.1 (release date: 1 August 2016), Version 4.2 (release date: 19 July 2017), Version 4.3 (release date: 16 July 2018), Version
4.4 (release date: 6 June 2019), Version 5.0 (release date: 28 June 2020), and Version 5.1 (release date: 22 November 2021). The forecast domain of the CMA-MESO model covers 17°–50°N, 102°–135°E before June 2019 and the whole China region (10°–60°N, 70°–145°E) thereafter with a horizontal grid spacing of 0.03° (Yu et al., 2018; Shen et al., 2020; Ma et al., 2022; Huang et al., 2022). The high horizontal resolution facilitates the comparison with point observations and minimizes the differences in altitude between the CMA-MESO grid points and the observation sites. There are 2501×1671
grid points in the north-south and east-west directions, respectively. A total of eight model cycle forecasts are run per day. The model, which has a 3Dvar assimilation system, adopts the analysis fields of the National Center for Environmental





Prediction Global Forecast System (NCEP/GFS) as the initial background field only at 00:00 UTC and 12:00 UTC, while the 3-h forecast fields of the previous cycle are used at all other cycle runs. The lateral boundary condition adopts the NCEP forecast field. Besides each of cycles running a 36-h integration at 00, 03, 06, 09, 12, 15, 18, 21UTC per day before 22 July 2021 (Ma et al., 2022), two cycles running a 72-h integration at 00UTC and 12UTC thereafter.

Given that the impact of the spin-up process on the forecast accuracy can be alleviated after 12-h operation of the model, the later 24-h (i.e., 12–36 h) instantaneous DnLWI predictions of the CMA-MESO model were chosen to compare with the corresponding in-situ measurements of the DnLWI. Apart from the DnLWI, ten meteorological variables predicted by the CMA-MESO were also adopted to illustrate how the meteorological factors influence the prediction deviation of the DnLWI from the CMA-MESO model (PDLR), which is the difference between the modelled and observed downward long-wave irradiances ($DnLWI_{meso}$–$DnLWI_{obs}$). The meteorological variables consist of the air temperature at 2-m height ($T_a$; unit: ℃), the surface pressure ($P_s$; unit: hPa), the relative humidity at 2-m height (RH; unit: %), the total column precipitable water vapour (PWV; unit: kg m$^{-2}$), the surface planetary boundary layer height (PBLH; unit: m), the surface visibility (Vis; unit: km), the column high cloud cover (HCC; unit: %), the column medium cloud cover (MCC; unit: %), the column low cloud cover (LCC; unit: %), and the total column integrated cloud ice (ICI; unit: kg m$^{-2}$). Furthermore, a total of 88 temperature (unit: K) and relative humidity (unit: %) profiles predicted by the CMA-MESO model were taken as input parameters into the MODTRAN model to simulate the spectral DnLWIs reaching the surface.

## 2.3 Methods

### 2.3.1 Spatiotemporal matching approach

As a prerequisite for the inter-comparison between the modelled and observed DnLWI, precise spatiotemporal matching is a key issue to solve the discrepancies arising from the spatial mismatch between the predicted gridded values and station observations as well as the time divergences caused by the hourly instantaneous prediction and the 1-min DnLWI measurements from the pyrgeometer. To this end, the DnLWI predictions of the CMA-MESO at the four grid points surrounding the validation site were interpolated to the coordinate of the site using the inverse distance weighting method. At the same time, ten 1-min DnLWI measurements centred at the punctual hour of the prediction were averaged to represent the observed DnLWI corresponding to the instantaneous prediction of the DnLWI.

### 2.3.2 Evaluation methods

Five statistical indicators, i.e., the mean bias error (MBE), the relative MBE (rMBE), the root-mean-square error (RMSE), the relative RMSE (rRMSE), and the correlation coefficient ($r$) are utilized in this study to evaluate the performance of the RRTM scheme in the CMA-MESO on the basis of in-situ DnLWI measurements. The MBE is an indicator to assess whether the prediction overestimates (positive value) or underestimates (negative value) in comparison with the measurements. The RMSE is an indicator used to account for the average magnitude of the errors, and the correlation coefficient $r$ reflects the



linear agreement between the observed and modelled variables (Gubler et al., 2012; Zhou et al., 2021). Definitions of these indicators are as follows:

$$\text{MBE} = \frac{1}{n}\sum_{i=1}^{n}(E_{m,i} - E_{o,i}),\tag{1}$$

$$\text{rMBE} = \frac{1}{n}\sum_{i=1}^{n}\frac{(E_{m,i} - E_{o,i})}{E_{o,i}} \times 100\%,\tag{2}$$

$$\text{RMSE} = \sqrt{\frac{1}{n}\sum_{i=1}^{n}(E_{m,i} - E_{o,i})^2},\tag{3}$$

$$\text{rRMSE} = \frac{\text{RMSE}}{\overline{E}_o} \times 100\%,\tag{4}$$

$$r = \frac{\sum_{i=1}^{n}(E_{m,i} - \overline{E}_m)(E_{o,i} - \overline{E}_o)}{\sqrt{\sum_{i=1}^{n}(E_{m,i} - \overline{E}_m)^2}\sqrt{\sum_{i=1}^{n}(E_{o,i} - \overline{E}_o)^2}},\tag{5}$$

where $E_{m,i}$ and $E_{o,i}$ represent the $i$ th values of the DnLWI modelled by the CMA-MESO and observed by the pyrgeometer, respectively. $\overline{E}_m$ and $\overline{E}_o$ are the mean values of the $E_{m,i}$ and $E_{o,i}$, respectively; $n$ is the sample number.

The Taylor diagram, in which the standard deviation (SD), RMSE, and $r$ between the test and reference are plotted within the same diagram (Talor, 2001), is adopted to evaluate the performance of the CMA-MESO and MODTRAN in predicting the DnLWIs against the observations. A Taylor skill score (TSS) is adopted to assess the prediction skill of the models in this study. The TSS is defined as (Hirota et al., 2011):

$$\text{TSS} = \frac{(1+r)^4}{4(\text{SDR} + 1/\text{SDR})^2}.\tag{6}$$

where $r$ is the correlation coefficient between the modelled and observed DnLWIs, and SDR is the ratio of the SD in the models against that of the observations. A higher TSS value indicates better simulation skills (Yang et al., 2019).

In addition, the partial least squares (PLS) method is applied in this study to investigate the potential influencing factors that result in the PDLR of the CMA-MESO model. The PLS method has been proven to have merits in resolving the multicollinearity among the dependent variable and its impact factors (McIntosh et al., 1996; Quan et al., 2023). In the PLS, values of the variable importance in projection (VIP) are adopted to identify the roles of the independent variables in interpreting the dependent variable (Jia et al., 2017).




## 3 Results and discussion

### 3.1 Performance of the CMA-MESO model in forecasting the DnLWI

#### 3.1.1 Comparison of the DnLWIs as observed and forecasted by the CMA-MESO model

Figure 1a shows a scatter density plot (SDP) of the DnLWI predicted by the CMA-MESO model versus corresponding measurements at 42 validation sites in China. The comparison was carried out based on the dataset described in Section 2, which consists of 587,158 pairs of hourly instantaneous observations and predictions of the DnLWI from July 2016 to December 2022. Thanks to its capability to plot scatter points and kernel density estimations (KDEs) in the same plot, the SDP plot can identify both the distribution of data points and their agglomeration degree (Jones and Sheather, 1991). It can be seen from Fig. 1a that the DnLWI predictions from the CMA-MESO model are reasonably consistent with those observed at the sites ($r$= 0.937) despite a tendency to underestimate the DnLWI (e.g., with an MBE of −6.6 W m$^{-2}$ and an rMBE of − 2.0%). The negative biases found in the CMA-MESO modelled DnLWIs were in line with the previous studies in the general circulation models (GCMs), in which the predictions of the DnLWIs generally lower than the observations in the range of 2 W m$^{-2}$ to 14 W m$^{-2}$ (Wild et al., 2001). On the other hand, the RMSE and rRMSE of the DnLWIs predicted by the CMA-MESO model in summer were 24.9 W m$^{-2}$ and 6.3%, respectively. As pointed out by Berdahl and Fromberg (1982), the sky's thermal radiation at a mid-latitude site during the summer is ~ 400 W m$^{-2}$, and a 5 per cent error in measurement or the estimation of this radiation (which is difficult to achieve) represents 20 W m$^{-2}$. Thereby, it is reasonable to believe that the DnLWIs predicted by the CMA-MESO model were comparable to those obtained at the sites. Moreover, most of the data points in Fig. 1a closely gather around 1:1 line (with the density values > 0.07), which means they have small RMSEs.

Figure 1b displays a histogram of the PDLR, which complies with the Gaussian distribution very well except for a slightly negative skewness. Moreover, the maximum frequency of the PDLR located in the interval of −10 to 0 W m$^{-2}$.

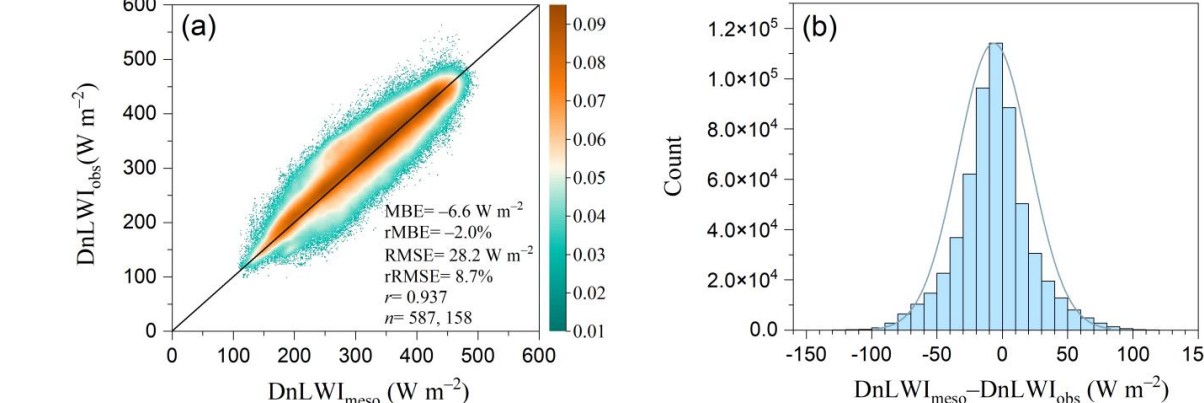

**Figure 1.** **(a)** Scatter density plot of the DnLWIs observed at the validation sites (DnLWI$_{obs}$) against those predicted by the CMA-MESO model (DnLWI$_{meso}$) under all conditions. **(b)** Histogram of the PDLR (DnLWI$_{meso}$− DnLWI$_{obs}$).The black thick line in (a) denotes the 1:1 line, and the blue curve in (b) is the Gaussian curve.





To further elucidate the performance of the CMA-MESO in predicting the DnLWI under various circumstances, the statistical results for all samples and the samples under different sky conditions, moistures, seasons, altitudes, climatic zones, and model versions were computed and listed in Table 3. A total of three sky conditions were defined in terms of the model output total cloud cover (TCC) according to the criteria presented by Morcrette (2002b), i.e., cloudless (TCC < 1%), partly

cloudy (1% ≤ TCC < 99%), and overcast (TCC ≥ 99%). With reference to the standard of Wolf and Gutman (2000), the humidity conditions of the atmosphere were classified as dry (PWV < 2.5 kg m$^{-2}$), semi-humid (2.5 ≤ PWV < 40 kg m$^{-2}$), and humid (PWV ≥ 40 kg m$^{-2}$) in terms of the PWV data output from the CMA-MESO model. Since the cloudiness and relative humidity were not included in the output of the model until the release of the CMA-MESO version 5.0, the total numbers of data pairs selected to identify the sky and humidity cases were 401,286 rather than 587,158 as for the other

categories. Moreover, we also categorized the sites into three types according to their altitudes, i.e., low altitude stations (< 500 m), medium altitude stations (500–1500 m), and high altitude stations (≥ 1500 m).

It can be seen from Table 3 that the CMA-MESO model commonly underestimates the DnLWI compared to the observations when the sky is cloudless (with an MBE of –15 W m$^{-2}$ and an rMBE of –5.2%) or partly cloudy (with an MBE of –10.3 W m$^{-2}$ and an rMBE of –3.1%), but overestimates it under overcast conditions (with an MBE of 10.3 W m$^{-2}$ and an

rMBE of 3.1%). The RMSEs were 26.5, 31.0, and 29.8 W m$^{-2}$, and the rRMSEs were 9.1%, 9.3%, and 8.9% under cloudless, partly cloudy, and overcast conditions, respectively. The correlation coefficients were greater than 0.920 for all sky conditions. In a clean atmosphere without clouds or aerosols, energy emitted from the Earth is largely absorbed by carbon dioxide, water vapour, ozone, and other trace gases in the atmosphere. The trapping of thermal infrared radiation by atmospheric gases is a typical characteristic of the atmosphere and is therefore called the *atmospheric effect* or *greenhouse*

*effect* (Liou, 2002). Clouds, which generally consist of water vapour, water droplets, or ice crystals, absorb thermal radiation very strongly and radiate similar to a black body in the infrared range (Heitor et al., 1991; Duarte et al., 2006; Li, et al., 2017; Khorsandi et al., 2023; Yang et al., 2023).

**Table 3.** Statistics between the DnLWIs observed and predicted by the CMA-MESO model under various conditions.

| | Type | MBE (W m$^{-2}$) | rMBE (%) | RMSE (W m$^{-2}$) | rRMSE (%) | r | Samples |
|---|---|---|---|---|---|---|---|
| Sky | cloudless | −15.0 | −5.2 | 26.5 | 9.1 | 0.951 | 143, 680 |
| | partly cloudy | −10.3 | −3.1 | 31.0 | 9.3 | 0.929 | 171, 393 |
| | overcast | 10.3 | 3.1 | 29.8 | 8.9 | 0.920 | 86, 208 |
| Humidity | dry | −9.3 | −5.3 | 23.8 | 13.0 | 0.699 | 18, 967 |
| | semi-humid | −8.3 | −2.7 | 30.5 | 9.8 | 0.891 | 330, 576 |
| | humid | −1.9 | −0.5 | 21.7 | 5.1 | 0.614 | 51, 738 |
| Season | spring | −7.9 | −2.4 | 28.8 | 8.9 | 0.894 | 128, 115 |
| | summer | −4.3 | −1.1 | 24.9 | 6.3 | 0.855 | 148, 180 |
| | autumn | −6.3 | −2.0 | 27.7 | 8.7 | 0.920 | 170, 448 |
| | winter | −8.2 | −3.2 | 31.5 | 12.4 | 0.887 | 140, 415 |





| | | | | | | | |
|---|---|---|---|---|---|---|---|
| Altitude | low | –6.9 | –2.0 | 26.5 | 7.8 | 0.946 | 335, 887 |
| | medium | –5.1 | –1.7 | 30.4 | 10.0 | 0.922 | 196, 573 |
| | high | –9.9 | –3.4 | 30.4 | 10.3 | 0.890 | 54, 698 |
| Climatic zone | TPCC | –2.9 | –1.0 | 30.0 | 10.4 | 0.914 | 164, 031 |
| | TPMC | –7.6 | –2.6 | 26.1 | 8.7 | 0.950 | 135, 669 |
| | STMC | –8.4 | –2.3 | 28.1 | 7.8 | 0.910 | 252, 976 |
| | TRMC | –6.0 | –1.5 | 25.4 | 6.3 | 0.801 | 12, 093 |
| | PLTC | –6.5 | –2.5 | 30.0 | 11.5 | 0.886 | 22, 389 |
| Version | MESO v4.1 | –6.8 | –2.0 | 26.4 | 7.6 | 0.949 | 53, 288 |
| | MESO v4.2 | –8.6 | –2.7 | 27.5 | 8.7 | 0.948 | 36, 582 |
| | MESO v4.3 | –0.3 | –0.1 | 24.7 | 7.4 | 0.952 | 38, 984 |
| | MESO v4.4 | –2.6 | –0.8 | 26.0 | 7.9 | 0.938 | 55, 203 |
| | MESO v5.0 | –11.3 | –3.2 | 29.1 | 8.3 | 0.926 | 116, 206 |
| | MESO v5.1 | –6.0 | –2.0 | 29.2 | 9.5 | 0.929 | 285, 455 |

The performance of the RRTM scheme in the CMA-MESO model improves with the increasing humidity. For instance, the MBEs (rMBEs) improved remarkably from –9.3 W m$^{-2}$ (–5.3%) under dry conditions to –1.9 W m$^{-2}$ (–0.5%) under humid conditions (Table 3). Meanwhile, the smallest rRMSE (5.1%) under humid conditions exhibited that the RRTM scheme was remarkably stable when the atmosphere was wet. In the infrared spectrum, the major absorbers in the clear-sky atmosphere are $H_2O$, $CO_2$, and $O_3$, thus, the rise of the water vapour content can dwarf the effects of the $CO_2$ and $O_3$ on the prediction of the DnLWI.

In summer, the temperature and humidity reach the maximum of the year, which resulted in the best performance of the RRTM scheme in the CMA-MESO model, i.e., the MBE, rMBE, RMSE, rRMSE were –4.3 W m$^{-2}$, –1.1%, 24.9 W m$^{-2}$, and 6.3%, respectively. On the contrary, the largest uncertainty of the RRTM scheme occurred in winter when the MBE, rMBE, RMSE, rRMSE were –8.2 W m$^{-2}$, –3.2%, 31.5 W m$^{-2}$, and 12.4%, respectively (Table 3).

Note that the best performance of the RRTM scheme in the CMA-MESO model appeared at medium altitude sites rather than at low or high altitude sites, which may be affected by the temperature, humidity, clouds, atmospheric pollutants, as well as the topography of the stations.

Among the five climatic zones, the DnLWI predicted by the CMA-MESO model has the smallest bias (with an MBE of –2.9 W m$^{-2}$ and an rMBE of –1.0%) in the TPCC zone. Nevertheless, large biases occurred in the TPMC zone, in which the MBE and rMBE were –7.6 W m$^{-2}$ and –2.6%, respectively. It is worth noting that the minimum RMSE (25.4 W m$^{-2}$) and rRMSE (6.3%) appeared in the TRMC zone may be attributed to the abundant water vapour in the atmosphere in this climatic zone. However, the CMA-MESO showed the most significant fluctuation (with an RMSE of 30.0 W m$^{-2}$ and an rRMSE of 11.5%) in predicting the DnLWI in the PLTC zone. The cold and dry air over the Qinghai-Tibet plateau brought a





challenge to the accurate prediction of the DnLWI from the CMA-MESO model. In addition, the most obvious MBE ($-8.4$ W m$^{-2}$) and relatively small rMBE ($-2.3\%$) of the DnLWI prediction in the STMC zone were probably caused by monsoon
activity, which usually brings abundant moisture into the atmosphere.

All versions of the CMA-MESO model underestimated the DnLWI to some extent. For instance, the DnLWIs were slightly underestimated in the version 4.3 (with an rMBE of $-0.1\%$) and version 4.4 (with an rMBE of $-0.8\%$). The version 4.1 and 5.1 had the same rMBE ($-2.0\%$), while the other two versions (i.e., the version 4.2 and the version 5.0) underestimated the DnLWI about $-3\%$. On the other hand, the largest uncertainty of the DnLWI predictions occurred in the
280 version 5.0, in which the MBE, rMBE, and the RMSE were $-11.3$ W m$^{-2}$, $-3.2\%$, and 29.1 W m$^{-2}$, respectively.

### 3.1.2 Diurnal variation of the DnLWIs observed and forecasted by the CMA-MESO model

Figure 2a showed the diurnal variation of the hourly DnLWIs observed and predicted by the CMA-MESO model, in which the DnLWIs reached their peak values in the afternoon (06:00–07:00 UTC) and minimum values in the early morning (~ 23:00 UTC). Notable underestimates of the DnLWI (rMBEs range from $-2.1\%$ to $-4.0\%$) predicted by the CMA-MESO
model appeared in daytime (00:00–12:00 UTC). During daytime, the diurnal variation of the convective planetary boundary layer (PBL) alters the stability of the lower layers of the atmosphere as well as the effective emissivity (Barbaro et al., 2010). Thereby, the discrepancy associated with the misrepresentation of the PBL during daytime might lead to the underestimation of the DnLWIs predicted by the CMA-MESO during the day time.

The DnLWIs observed were generally lower than those predicted by the CMA-MESO model under the cloudless and
290 partly cloudy conditions (Fig. 2b and 2c), but higher than those under the overcast conditions (Fig. 2d). In addition, the amplitude of diurnal variation of the DnLWIs predicted by the CMA-MESO model was smoother under overcast conditions than that under cloudless or partly cloudy conditions. The presence of clouds not only increases the intensity of downward long-wave radiation but also decreases its diurnal cycle amplitude because clouds can emit in the atmospheric window and the cloud-base temperature does not vary much during the day (Barbaro et al., 2010).
With the increase of atmospheric humidity, the diurnal cycles of the DnLWIs observed and predicted by the CMA-MESO got closer and closer (Fig. 2e–2g). In contrast, an apparent underestimation of the predicted DnLWI under dry conditions during the daytime (rMBE= $-7.6\%$) was mostly come from the insufficient consideration of the PBL in the CMA-MESO model.

The CMA-MESO model considerably underestimated the DnLWIs (with an rMBE of $-3\%$) in winter (Fig. 2k), which
may be related to the insufficient prediction ability of the RRTM scheme under extremely cold and dry conditions. The DnLWIs predicted by the CMA-MESO model, whereas, matched the observations very well at night in spring, summer, and autumn (Fig. 2h–2j). Furthermore, the discrepancies of the DnLWI predicted during daytime in summer (with an rMBE of $-2\%$) were less than those in autumn (with an rMBE of $-3\%$) and in spring (with an rMBEs of $-34\%$) owing to the plentiful water vapour and more clouds in summer.

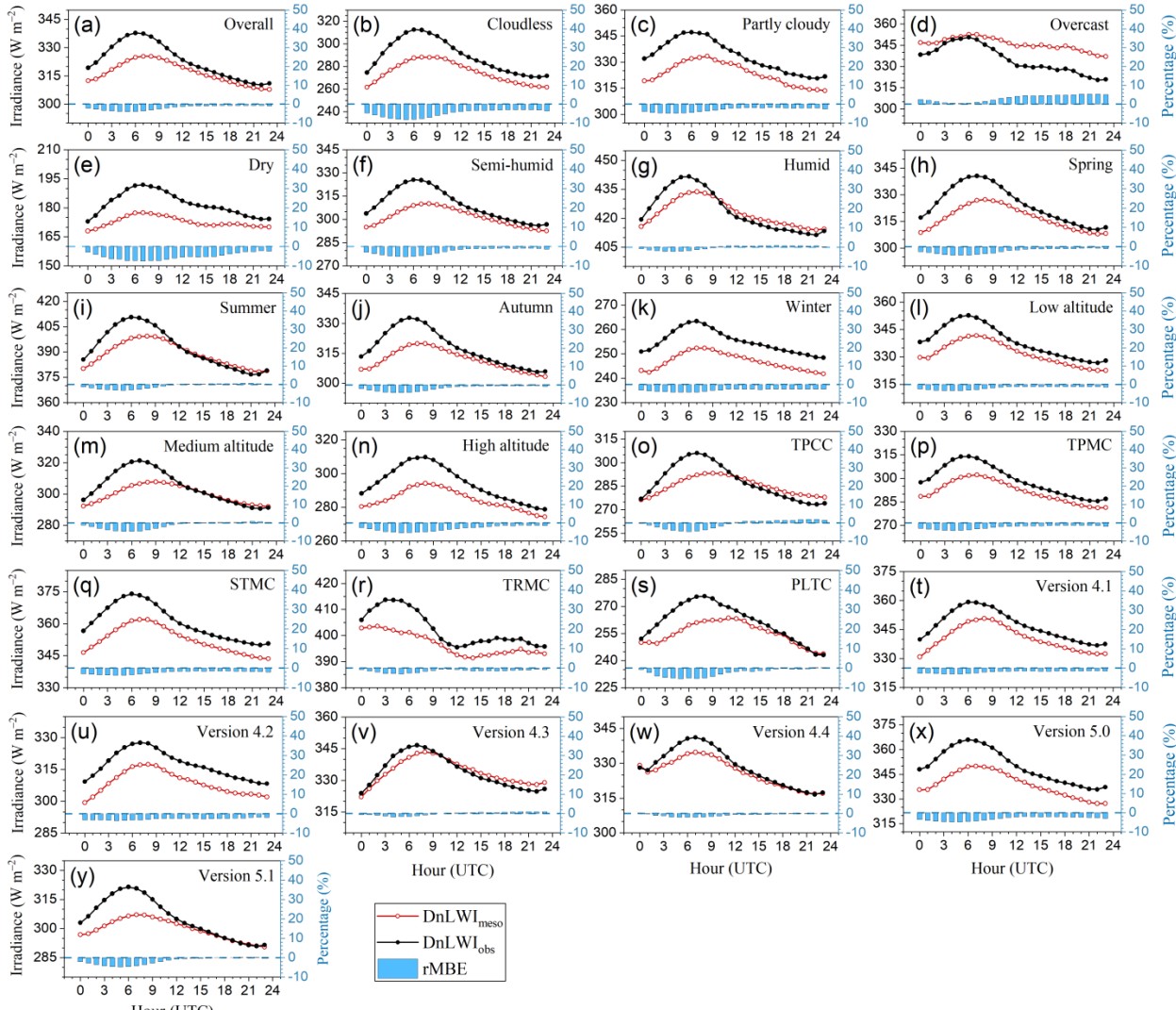

**Figure 2.** Diurnal variation of mean hourly DnLWI observations (denoted with the black lines with dots) and predictions from all versions of the CMA-MESO model (denoted with the red lines with circles) **(a)** for overall samples; under **(b)** cloudless, **(c)** partly cloudy, **(d)** overcast, **(e)** dry, **(f)** semi-humid, and **(g)** humid conditions; in **(h)** spring, **(i)** summer, **(j)** autumn, and **(k)** winter; at the sites located in **(l)** low altitudes, **(m)** medium altitudes, and **(n)** high altitudes; as well as in the climatic zones of **(o)** the TPCC, **(p)** the TPMC, **(q)** the STMC, **(r)** the TRMC, and **(s)** the PLTC. Furthermore, the mean hourly DnLWIs observed and predicted by various versions of the CMA-MESO model were also calculated and plotted for **(t)** the version 4.1, **(u)** the version 4.2, **(v)** the version 4.3, **(w)** the version 4.4, **(x)** the version 5.0, and **(y)** the version 5.1, respectively. In addition, the blue columns in Fig. 2 represented the related rMBEs of the DnLWIs predicted by the CMA-MESO model.

It can be seen from the Fig. 2l–n, the hourly DnLWIs predicted by the CMA-MESO model were less than the observations for all altitudes except for the medium-altitude during the night (13:00–23:00 UTC), in which the DnLWIs predictions fit the observations very well. The temperature, air humidity, and clouds over the stations at different altitudes





depend not only on the elevations of the sites but also on their geographical locations, which complicates the relationships between the hourly DnLWIs observed and those predicted by the CMA-MESO model.

The diurnal variations of the hourly DnLWIs in the climatic zones were modulated by the air temperature and water vapour in the atmosphere. For example, the maximum value of the DnLWI observed in the TRMC (415 W m$^{-2}$) was 139 W m$^{-2}$ higher than the one in the PLTC (276 W m$^{-2}$), which arose from the differences in the air temperature and water vapour content between two climatic zones (Fig. 2r and 2s). Moreover, the hourly DnLWIs predicted by the CMA-MESO model were less than those observed in all cases except those in the TPCC during the night (Fig. 2o).

After comparing the performances of various versions of the CMA-MESO model, it was found that the DnLWIs predicted by the version 4.3 and 4.4 of the CMA-MESO model agreed with the observations to a large extent (Fig. 2v and 2w). However, other three versions (i.e., version 4.1, 4.2, and 5.0) of the CMA-MESO model underestimated the DnLWIs throughout the day. It is worth noting that the version 5.1 accurately predicted the hourly DnLWIs at night but remarkably underestimated them in the daytime. As mentioned above, the reliability of DnLWI predicted by the CMA-MESO model depends not only on the RRTM scheme employed in the model but also on several key physical processes (e.g., the cloud microphysics, surface layer, planetary boundary layer, turbulence, etc.), as well as the data assimilation system and dynamic process adopted in the model (Xu et al., 2008).

### 3.2 Possible impacts on the RRTM in the CMA-MESO

### 3.2.1 Analysis of the impact of different factors on the deviation of the DnLWI forecast

To reveal the possible influence of the different meteorological elements on the PDLR in the CMA-MESO model, the PDLR against individual meteorological variables predicted by the CMA-MESO model were plotted in Fig. 3. The PDLR negatively correlates with $T_a$, $P_s$, and PBLH, and the corresponding correlation coefficients were of –0.055, –0.044, and –0.183, respectively (Fig. 3a, 3b, and 3e). On the other hand, with increasing temperature and surface pressure, the amplitude of the PDLR slightly decreased, which implies that the CMA-MESO model could yield relatively stable DnLWI over the warm and low altitude areas. Figure 3e indicates that the amplitude of the PDLR declined with the increase of the PBLH, which may be related to the atmospheric emission in the PBL. Previous investigations pointed out that as much as 60% (90%) of atmospheric emission is derived from the atmosphere within the first 100 m (1 km) under clear-sky conditions, and more than 90% originates from within the first 1 km between the ground and the bottom of the cloud if the sky is overcast (Ohmura, 2001). Moreover, the downward long-wave radiation measurements at the surface also contain enough information about the PBL to infer the thermal vertical structure of the lower layers by comparing broadband LW measurements and LW in the atmospheric window (Gröbner et al., 2009).



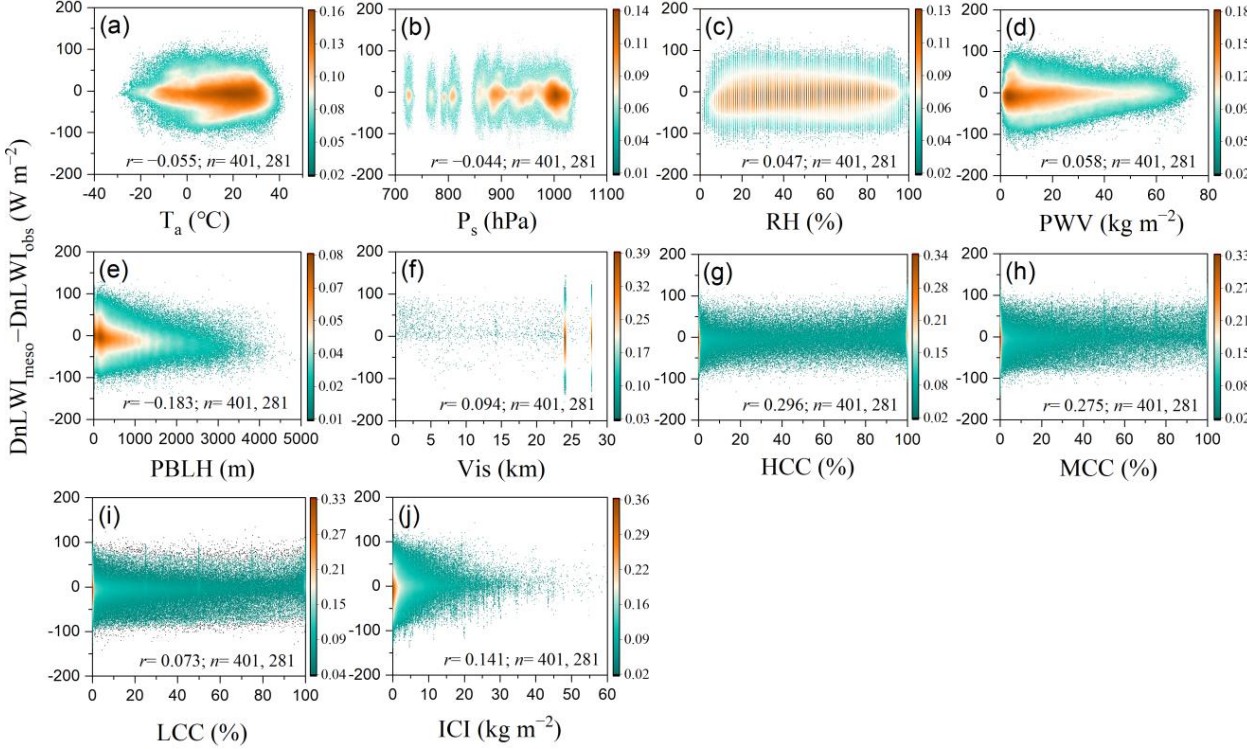

**Figure 3.** Scatter density plots of the hourly PDLR as a function of concurrent predictions of ten meteorological variables from the CMA-MESO model such as **(a)** air temperature $T_a$, **(b)** surface pressure $P_s$, **(c)** relative humidity RH , **(d)** total column integrated water vapour PWV, **(e)** planetary boundary layer height PBLH, **(f)** visibility Vis, **(g)** high cloud cover HCC, **(h)** medium cloud cover MCC, **(i)** low cloud cover LCC, and **(j)** column integrate cloud ICI**.**

Nevertheless, positive correlation coefficients existed between the PDLR and the RH (0.047), PWV (0.058), Vis (0.094), HCC (0.296), MCC (0.275), LCC (0.073), and ICI (0.141), respectively (Fig. 3c–3d, 3f–3j). All correlation coefficients were significant at the 95% confident level. Note that, the RH and PWV represent the air humidity near the surface and in the whole atmosphere, respectively. The more water vapour in the atmosphere, the more downward long-wave radiation is emitted (e.g., Ohmura, 2001). As an indirect indicator represents the transparency of the atmosphere, the Vis under clear sky is proven to have a good negative correlation with the aerosol optical depth (Kaufman and Fraser, 1983; Wu et al., 2014) and the vertical distribution of aerosol particle number density (Elterman, 1970). Aerosols can heat the Earth's surface through enhancing the downwelling long-wave radiation, which depends on the types, concentration, and height of the aerosol. Unfortunately, due to the lack of chemical transmission module in the CMA-MESO model, the effect of aerosols is not abundantly considered during predicting the DnLWI, which may result in a small correlation coefficient (0.094) between the DnLWI and Vis (Fig. 3f).

Despite the correlations between the PDLR and ten individual influencing factors were described in detail above, it is well know that multicollinearity exists among those influencing factors (Quan et al., 2023). In this section, we further explore the relationships between the PDLR and influencing factors by using the PLS analysis method. The merit of the PLS analysis is





to resolve the collinearity exists among the independent variables by means of comparison of their VIP values. Under all sky conditions, the VIP values of the HCC, MCC, PBLH, and ICI were 1.82, 1.69, 1.15, and 1.03, while the other six factors (i.e., the Vis, RH, LCC, PWV, $T_a$, and $P_s$) had VIP values less than 0.8 (Fig. 4a). Under clear skies, the VIP values of the three factors were greater than 0.8, i.e., $P_s$ (1.34), Vis (1.27), and RH (1.24), while the other three factors had VIP values smaller than 0.8 (Fig. 4b). The variables with the VIP values less than 0.8 were generally considered to have minor contribution to

explaining the PDLR, i.e., the variable with a VIP less 0.8 is considered as either intrinsically less important to the dependent variable or important to the dependent variable but it is well predicted by the CMA-MESO and its slight prediction uncertain has a small contribution to the independent variable.   Effect of the cloudiness on the DnLWI is essentially complicated because it relies on the cloud cover as well as the cloud type, cloud base height, cloud thickness, etc. (Duarte et al., 2006; Sugita and Brutsaert, 1993).

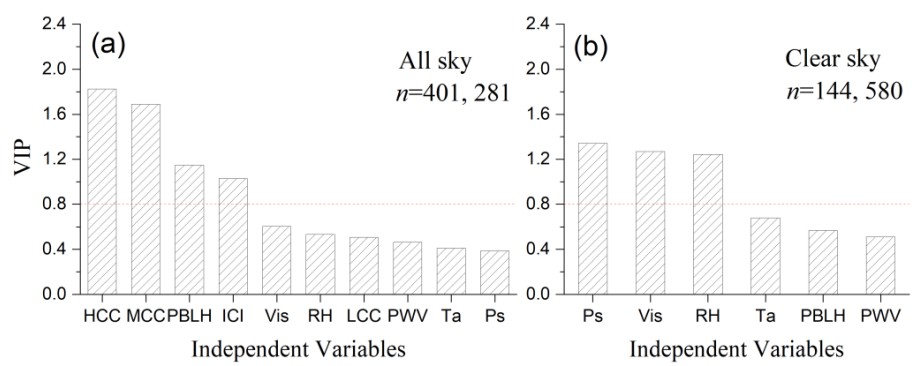

**Figure 4.** VIP histograms of the impact factors related to the forecasting deviations of the DnLWI under **(a)** all-sky conditions, and **(b)** clear-sky conditions. Red dotted lines denote the thresholds.

To elucidate the accuracy of the cloud cover predicted by the CMA-MESO model, cloud cover differences (CCDs) of the

TCC, HCC, MCC, and LCC were calculated by subtracting the monthly grid cloud cover data of the ERA5 reanalysis data in 2022 from the counterparts predicted by the CMA-MESO model. The ERA5 is an atmospheric reanalysis produced by the ECMWF (Hersbach et al., 2020), and its monthly data were gridded to a regular lat-lon grid of 0.25 degrees for the reanalysis (https://cds.climate.copernicus.eu/cdsapp#!/dataset/reanalysis-era5-single-levels-monthly-means?tab=form; last access: 23 February 2024). The results showed that the CMA-MESO model underestimated the TCC and MCC (with the CCDs less

than −10%) over most regions of China. The CCDs of the HCC were less than −5% over whole China except for the Qinhai-Tibet Plateau, over which the CCDs were greater than 5%. However, the LCC predicted by the CMA-MESO model were relatively unbiased, i.e., the CCDs of the LCC ranged from −5% to 5% over most regions of China. On the other hand, the significant underestimation of the DnLWIs (−5.5% < rMBE ≤ −4.0%) occurred at 8 stations (i.e., YON, BS, GY, DL, YC, TY, SY, and JN), where the CCDs of all cloud types (i.e., the TCC, HCC, MCC, and LCC) were less than −5%. Thereby, it

can be inferred that the underestimation of the DnLWIs from the CMA-MESO model was may relate to its underestimation of the cloud cover (e.g., Kang et al., 2016).





### 3.2.2 Elucidation of the impact mechanisms using spectral irradiance simulations from the MODTRAN

To further elucidate the possible influencing mechanisms of the atmosphere and clouds on the DnLWI prediction, a total of 88 randomly selected temperature and relative humidity profiles produced by the CMA-MESO model under various conditions were taken as input parameters into the MODTRAN version 4.1 to simulate the DnLWI spectrum reaching the surface. Note that these profiles have a total of 33 vertical layers, in which the first 19 layers (i.e., 1000, 975, 950, 925, 900, 850, 800, 750, 700, 650, 600, 550, 500, 450, 400, 350, 300, 200, and 100 hPa) were derived from the outputs of the CMA-MESO model, and the remaining 14 layers (80, 60, 50, 40, 35, 30, 20, 10, 5, 2.5, 1.5, 0.5, 0.05, 0.01 hPa) were replaced with the corresponding profile data of the standard atmospheres, e.g., the middle latitude summer, the middle latitude winter, and the tropical atmosphere. Meanwhile, the annual mean ozone profile adopted in the CMA-MESO model was taken as the input into the MODTRAN model after interpolating to the 33 vertical layers. The default $CO_2$ density (330 ppmv) used in the CMA-MESO model was input into the MDDTRAN during the simulation of the spectral DnLWIs. Furthermore, the clouds predicted by the CMA-MESO model were also input into the MODTRAN model after some necessary transformations. For instance, the clouds were transferred into three types of cloud according to the height of the cloud base, i.e., high cloud (with cloud base pressure less than 440 hPa), medium cloud (with cloud base pressure less than 680 hPa but greater than 440 hp), and low cloud (with cloud base pressure greater than 1200 hPa). As one of the output simulation files of the MODTRAN model, a *Tape8* file consists of spectral irradiance data, in which the anisotropy of downwelling long-wave radiation is well taken into account. Subsequently, the spectral irradiance data from the *Tape8* file were spectrally integrated to produce the DnLWIs.

It can be seen from Fig. 5, the variability of the observed DnLWIs for 88 cases agreed well with the corresponding simulations from the CMA-MESO and the MODTRAN models, except for some underestimates appeared in the cases 14 to 19, 37 to 40, and 76 to 86, which may be caused by the unreliable input profiles and other parameters input to the models.

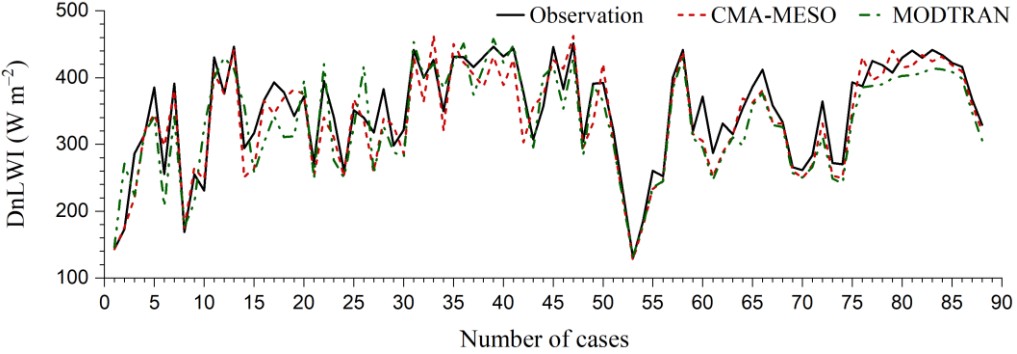

**Figure 5.** Series of the DnLWIs observed at stations for 88 randomly selected cases, as well as those predicted by the CMA-MESO and MODTRAN models.

The Taylor diagram of the DnLWIs simulated with the CMA-MESO model and MODTRAN model under all sky conditions indicated that the performance of the CMA-MESO was generally better than that of the MODTRAN model (Fig. 6a). For example, the correlation coefficients between the DnLWIs observed and predicted by the CMA-MESO and





MODTRAN model were 0.943 and 0.907, respectively. The root mean square errors were 28.8 and 36.8 W m$^{-2}$ for the CMA-MESO model and the MODTRAN model. The TSS value of the CMA-MESO model (0.893) was higher than that of the MODTRAN model (0.828). As shown in Fig. 6b, the CMA-MESO model (with a TSS of 0.923) performed slightly worse than the MODTRAN model (with a TSS of 0.949) under clear sky conditions. Previous investigators pointed out that large aerosol particles can scatter infrared radiation to some extent (Zhou and Savijärvi, 2014), which was taken into account in the MODTRAN model via the input visibility but not considered in the CMA-MESO model. In particular, the performance of the CMA-MESO (with a TSS of 0.732) was obviously worse than the MODTRAN model (with a TSS of 0.934) in simulating the DnLWIs under high and medium clouds (Fig. 6c), but was better (with a TSS of 0.883) than that of the MODTRAN model (with a TSS of 0.703) under low clouds (Fig. 6d). This phenomenon showed that the reliability of the DnLWIs predicted by the CMA-MESO model may be affected by the inappropriate input of the cloud types, especially the high and medium clouds.

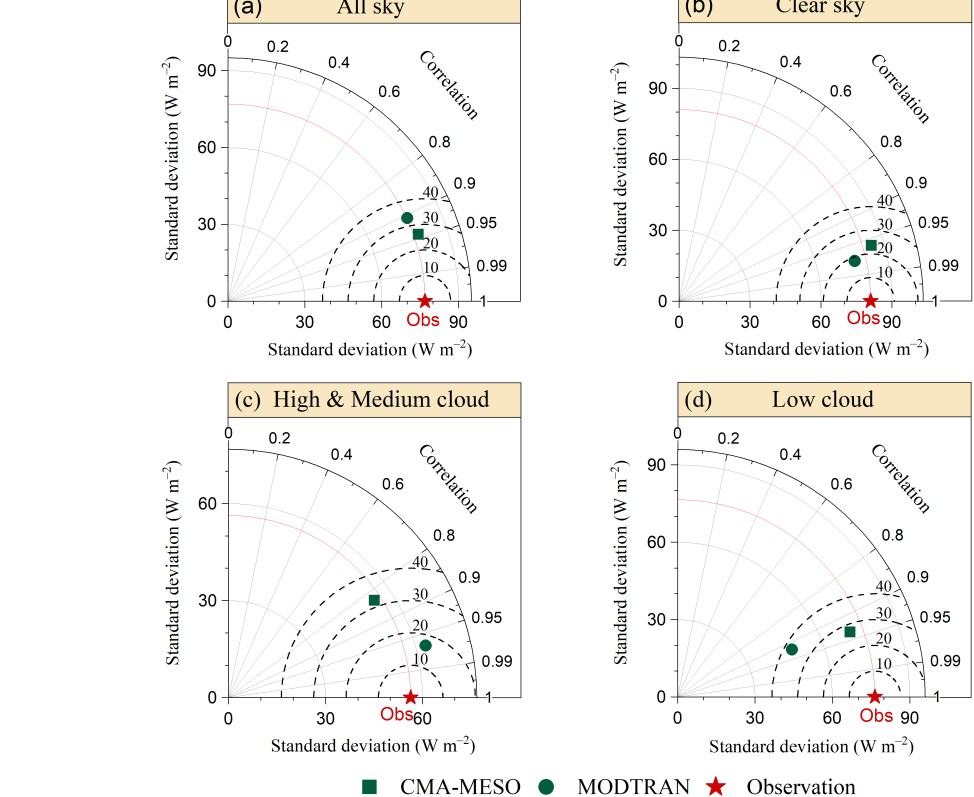

**Figure 6.** Taylor diagrams describe the RMSEs, standard deviations, and correlation coefficients between the observed DnLWIs and those predicted by the CMA-MESO model and MODTRAN model in the case of **(a)** all-sky, **(b)** clear-sky, **(c)** high and medium clouds, and **(d)** low clouds. The "Obs" is considered as the reference point, the closer the point of the model prediction to the "Obs", the better performing is the model.

As the CMA-MESO model merely output irradiance rather than spectral irradiance, the MODTRAN model was used along with the input atmospheric profiles at the XL station, which were five typical cases selected from the 88 cases, to yield the




irradiance spectrum. Table 4 indicates that the five cases at the XL site cover wide atmospheric states and sky conditions, in which the screen level temperature ranges from –20.2 to 29.6 ℃, and the PWV varies from 0.16 to 2.30 g cm$^{-2}$. The reason why uses the spectral irradiance is to facilitate detect the impacts of the atmospheric components and clouds on the DnLWI. Since the same atmospheric profiles were used as input parameters to both the CMA-MESO model and the MODTRAN
model, the comparisons between the DnLWIs predicted by the two models, thus, focused on the discrepancies between the RRTM scheme and the radiative transfer model.

**Table 4.** Basic descriptions of selected cases at XL site used to calculate the spectral DnLWIs with the MODTRAN model.

| No. | Time (UTC) | Sky condition | Cloud type | Cloud amount (%) | $T_a$ (℃) | PWV (g cm$^{-2}$) | $CO_2$ (ppmv) |
|---|---|---|---|---|---|---|---|
| 1 | 2020-12-18T11:00 | Clear sky | – | 0 | –20.2 | 0.16 | 330 |
| 2 | 2020-10-14T04:00 | Partly cloudy | High cloud | 9.2 | 1.5 | 0.60 | 330 |
| 3 | 2020-07-24T10:00 | Partly cloudy | Medium cloud | 21.4 | 29.6 | 1.97 | 330 |
| 4 | 2021-02-25T00:00 | Partly cloudy | Low cloud | 34.3 | –11.5 | 0.26 | 330 |
| 5 | 2020-08-27T07:00 | Overcast | High cloud | 100.0 | 26.1 | 2.30 | 330 |

Figure 7 shows the profiles of the temperature and the densities of the $H_2O$, $O_3$, and $CO_2$, which were transformed from the original input profiles of the temperature, relative humidity, and the $O_3$ density, as well as the near-surface $CO_2$
mixing ratio through the MODTRAN model. It can be seen from Fig. 7a that the temperature decreases with increasing altitude in the troposphere, and the tropopause height reaches at ~ 16 km in summer due to the strong convection in the atmosphere (case 3 and 5) while reaches at ~ 10 km in other seasons (Case 1, 2, and 4).

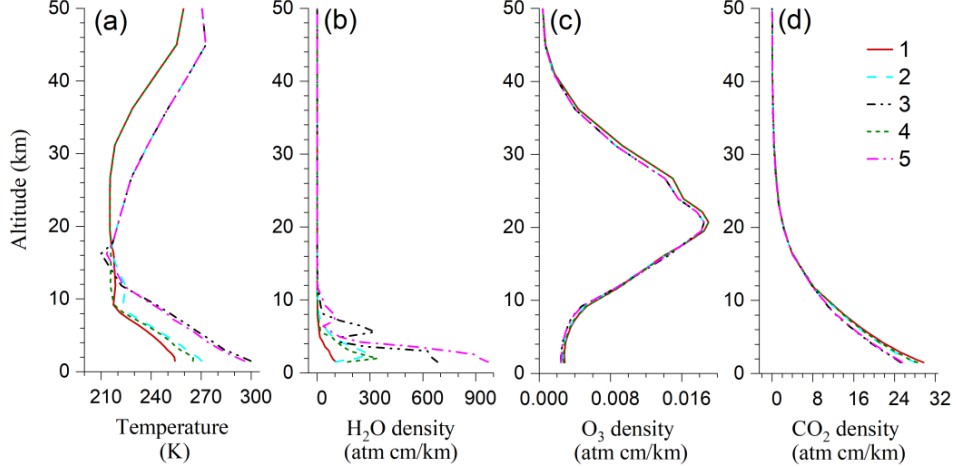

**Figure 7.** Atmospheric profiles of the **(a)** air temperature, the densities of **(b)** $H_2O$, **(c)** $O_3$, and **(d)** $CO_2$ for the five cases at XL site. All these profiles are modulated by the MODTRAN model in terms of the corresponding inputs.

The $H_2O$ density decreased rapidly with increasing height, e.g., though $H_2O$ densities at the lower atmosphere varied greatly among the different profiles, they closed to zero when the atmospheric altitude greater than 10 km (Fig. 7b). Not





that the same $O_3$ profile was used as input in the five cases, thereby, the transformed $O_3$ profiles were very close (Fig. 7c). In addition, the $CO_2$ profiles decreased exponentially with increasing height (Fig. 7d).

Figure 8 showed the spectral absorption curves over the spectral range of 3.0–30 μm, which were simulated by the
495 MODTRAN model for the five cases at XL site. Among them, the $\nu_2$ fundamental band at 6.25 μm is the most important vibrational-rotational band of water vapour. In the well-known "*thermal infrared window*" (8.3 to 12.5 μm), a moderately strong 9.6 μm band of ozone is contained. The attenuation of water vapour continuum in the 10 μm window remains a theoretical mystery, which may be caused by the accumulated absorption of the distant wings of water vapour lines (Liou, 2002). It is worth noting that the far infrared absorption band (17–33 μm) of water vapour, which is also named as a "*micro*
*window*" or a "*secondary atmospheric window*", plays an important role in the Earth energy balance and climate related issues (Marty et al., 2003; Gröbner et al., 2009). If the atmosphere is very dry (e.g., PWV=0.16 g cm$^{-2}$ in Fig. 8a), the far infrared band is almost semi-transparent (the mean absorptivity is about 0.5) and the associated downward irradiances mainly represent the colder temperatures. In contrast, the secondary atmospheric window is opaque when the highest concentration of atmospheric water vapour occurred (e.g., PWV=2.30 g cm$^{-2}$ in Fig. 8e).

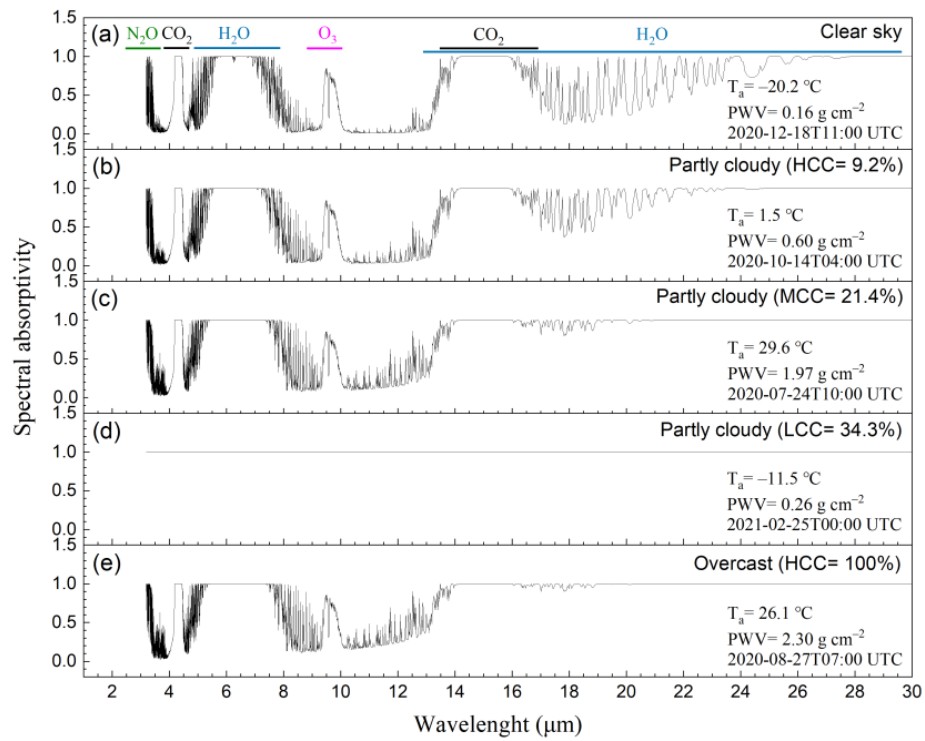

**Figure 8.** Spectral absorption curves of the atmosphere that simulated with the MODTRAN model along with atmospheric profiles at the XL site for **(a)** case 1, **(b)** case 2, **(c)** case 3, **(d)** case 4, and **(e)** case 5. All absorbing gases and their spectral absorption bands are also denoted.

As for $CO_2$, its vibration band centred at 15 μm is the main absorption band in the infrared region, which is critical for
atmospheric radiative heat exchange. Meanwhile, the 4.3 μm-band is a very narrow and strong absorption band of $CO_2$, in





which the solar radiation can be completely absorbed by the atmosphere above 20 km altitude and contributes little to the DnLWI (Liou, 2002).

In the presence of low clouds, the spectral absorptivity is close to 1.0 (Fig. 8d) because the low clouds consist of small water droplets and be identified as the radiatively black clouds (Liou, 2002). On the contrary, the high and medium clouds

(e.g., the cirrus clouds) are identified as the non-black clouds and introduce more uncertainties than the black clouds (e.g. the low-level stratiform clouds) in the NWP system during simulating the DnLWIs (Khorsandi et al., 2023).

Figure 9 illustrates the spectral DnLWIs reaching the surface at XL site for the five cases. The DnLWI is composed of two parts: one is the thermal radiation emitted by the atmosphere, which is mainly from the emission of water vapour and $CO_2$ within a layer of 1–2 km above the ground, and the other is the cloud thermal radiation, which is emitted by the clouds and

propagated to the ground through the atmospheric window (Liou, 2002; Marty et al., 2003; Yang et al., 2023). Thereby, the DnLWI mainly depends on the screen level temperature, but is also modulated by the vertical distributions of the air temperature and humidity, greenhouse gases (e.g., $CO_2$ and $O_3$), as well as clouds.

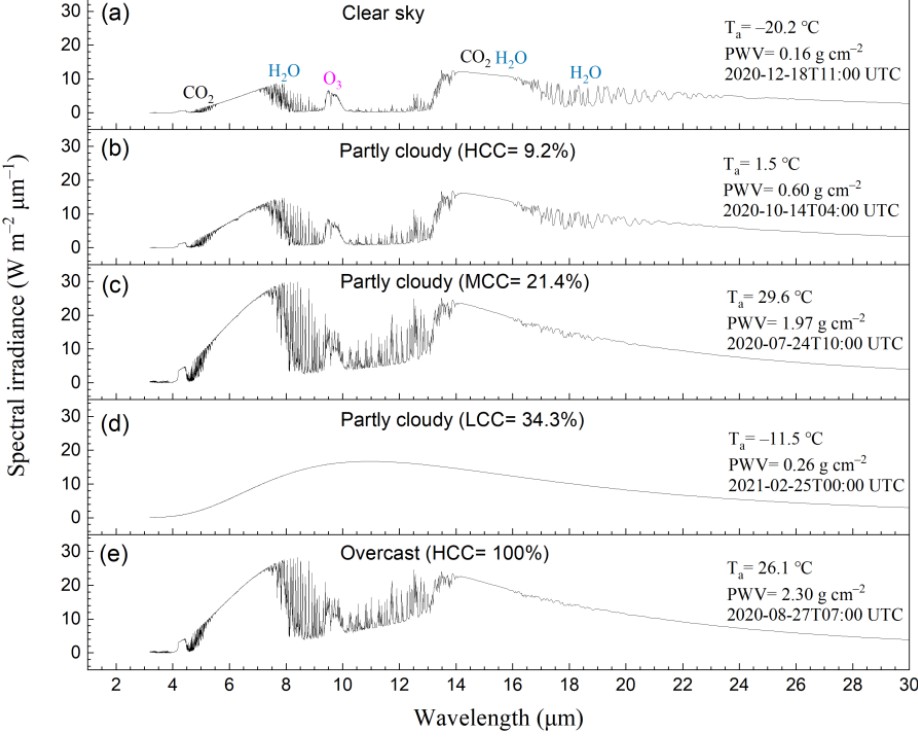

**Figure 9.** Same as Fig. 8 except for the spectral DnLWIs reaching the surface.

It can be seen from Fig. 9 that the amplitudes of the spectral DnLWI largely depend on the screen level temperature. For instance, the maximum spectral irradiances were 12.5 and 29.8 W $m^{-2}$ $\mu m^{-1}$ and the corresponding screen level temperatures were −20.2 and 29.6 ℃ in the case 1 and 3 (Fig. 9a and 9c), respectively. Meanwhile, water vapour significantly enhanced the thermal radiation emissions in the *thermal infrared window* and the *secondary atmospheric window* (Fig. 9e). Additionally, as the thermal emission of the greenhouse gases depends not only on the intensity of the gas





absorption line but also on their tiny amounts, the increasing water vapour, thus, can dwarf the relative contributions of $CO_2$ and $O_3$. Moreover, the effects of clouds on DnLWI rely on the cloud types as well as the cloud amounts. For instance, even a low cloud amount (34.3%) of the LCC can produce the spectral irradiance complies with the Planck law (Fig. 9d).

## 4 Summary and conclusions

In this study, the RRTM scheme in the CMA-MESO model was systematically evaluated on the basis of high time-
resolution observations of the DnLWI at 42 sites in China. As a whole, the CMA-MESO model yields satisfactory results, i.e., the DnLWI predictions are largely consistent with the observations with a correlation coefficient of 0.937. The CMA-MESO model, whereas, has a slight tendency to underestimate the DnLWI (e.g., with an MBE of −6.6 W m$^{-2}$ and an rMBE of −2.0%) under all conditions. It is worth noting that the DnLWIs predicted by the CMA-MESO model agreed well with the observations under warm and humid conditions but were significantly underestimated in the case of the cold and dry
atmosphere. This phenomenon partly arises from the intrinsic shortcoming of the RRTM scheme in the CMA-MESO model to insufficiently simulate the thermal emission of the cold and dry atmosphere, and partly relates to the unreasonably large values of the DnLWI observed by the pyrgeometers under extreme cold and dry synoptic conditions (Yang et al., 2023). With the increasing cloud cover, especially the low cloud cover, the DnLWIs predicted by the CMA-MESO model will change from underestimation to overestimation due to the strong thermal emission of the clouds.

Both the DnLWIs observed by the pyrgeometer and those predicted by the CMA-MESO model exhibit similar characteristics of the diurnal variation. Nevertheless, the hourly DnLWIs predicted by the CMA-MESO model are lesser than observations for all cases except the cases under overcast conditions. Moreover, the consistence between the predicted and observed DnLWI is better at night than that during the day, which may be related to the stable nocturnal boundary layer.

The VIP values of the PLS analysis indicate that the HCC, MCC, PBLH, and ICI are potential factors affecting the
accuracy of the DnLWI predicted by the CMA-MESO model under all sky conditions. The most important factors that potentially influence the accuracy of the DnLWI predictions are $P_s$, Vis, and RH under clear sky conditions.

Cross-comparing between the DnLWIs predicted by the MODTRAN model and the CMA-MESO model implies that the uncertainty of the CMA-MESO forecasting is likely arisen from the inaccuracy of the estimation of the high or medium clouds in the CMA-MESO model.

In the near future, several investigations related to the CMA-MESO model are expected to be carried out. One is to validate the cloud products predicted by the CMA-MESO model on the basis of in-situ observations of cloud properties because uncertainties in clouds can bring a large error in the DnLWIs predicted by the CMA-MESO model. The other is to further verify the shortwave scheme in the CMA-MESO model using the methods as presented in this study. At last, sensitivity tests of greenhouse gases (e.g., $CO_2$ and $O_3$) on the uncertain of the DnLWI prediction under clear sky conditions
are looking forward to be performed by virtue of the MODTRAN model.



*Code and data availability.*

The dataset used in this study is available online at https://doi.org/10.5281/zenodo.10971844 (Yang and Quan, 2024).

*Author contributions.*

JY contributed the ideas, processed the in-situ radiation data and the outputs of the CMA-MESO model, carried out the comparison analysis, and edited the manuscript. WQ designed the experiments, performed the PLS analysis and MOTRAN simulations, edited the manuscript, and provided the funding acquisitions. LZ, JH, and QC provided suggestions and comments on model data usage and comparison analysis method. MW contributed to improving the ideas and revised the manuscript.

*Competing interests.*

The contact author has declared that none of the authors has any competing interests.

*Disclaimer.*

Publisher's note: Copernicus Publications remains neutral with regard to jurisdictional claims made in the text, published maps, institutional affiliations, or any other geographical representation in this paper. While Copernicus Publications makes

every effort to include appropriate place names, the final responsibility lies with the authors.

*Acknowledgements.*

We appreciate Qifeng Lu, Jin Zhang, Fei Yu, and Yingjie Cui of the CEMC for providing valuable suggestions and introduction of the model versions. We would like to thank Yan Shi and Jie Liao of the National Meteorological Information Centre for offering assistance to process the raw data.

*Financial support.*

This research has been supported by the China Scholarship Council (grant no. 202205330024), the National Key Research and Development Program of China (grant no. 2017YFB0504002), and the Second Tibetan Plateau Scientific Expedition and Research (STEP) program (grant no. 2019QZKK0105).

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
