# Peer review of "Evaluation of radiation schemes in the CMA-MESO model using high time-resolution radiation measurements in China: I. Long-wave radiation"

_Geoscientific Model Development, 2024_

## Referee Comment (RC1)

General comments:

This paper provides a comprehensive evaluation of the CMA-MESO model's ability to predict downward long-wave irradiance (DnLWI) in China, using extensive high time-resolution in-situ measurements. The study is significant as it identifies the model's performance under various atmospheric conditions and highlights discrepancies particularly under overcast, dry, and cloudless scenarios. The authors also pinpoint the influence of cloud cover and the stable nocturnal boundary layer on the model's accuracy. This thorough assessment offers valuable insights for improving the radiation schemes in NWP models and underscores the importance of accurate cloud representation in enhancing model reliability. The manuscript is suitable for publication, once minor revisions are made.

The title of the manuscript, "Evaluation of radiation schemes in the CMA-MESO model using high time-resolution radiation measurements in China: I. Long-wave radiation," is somewhat misleading. The study exclusively assesses downward long-wave irradiance (DnLWI), without addressing other aspects such as downward long-wave irradiance in specific atmospheric layers or upward long-wave irradiance. The authors can find their suitable words to clarifying this in the title would better reflect the manuscript's focus and scope. Probably, for instance, change "Longwave radiation" to "surface downward Longwave radiation".

Specific comments:

1. Line 132: "quality control". The pygeometer measurements are only valid within certain spectral ranges. Did this 'quality control 'step consider the

response function of the pygeometer? To validate the LW radiation scheme using in-situ pygeometer measurements, signals of DnLWI predictions from the CMA-MESO that are beyond the pygeometer's spectral range must be filter out. Did this 'quality control 'step consider masking those signals?

2. Lines 161-162: " ….were taken as input parameters into the MODTRAN model " I wonder if the atmospheric profile inputs used to drive the RRTM in the CMA-MESO and MODTRAN are identical? So that comparisons between RRTM and MODTRAN are reasonable to be make.

3. Line 324-331: Figures 2u-y show performances of various versions of the CMA-MESO model. Are there any possible reasons explaining the different performance across various versions of the CMA-MESO model? I recommend the authors to added 2-3 sentences (or probably more sentences) to simply elucidate this somewhere in the text.

4. Line 400-406: " …… were replaced with ….. the middle latitude summer, the middle latitude winter…… " The 88 profiles predicted by the CMA-MESO are 3-h forecast field across 8 hours cycle per day. Right? The Total inputs for MODTRAN include 33 layers, this first 19 layers are from CMA-MESO, the remaining 14 layers are from standard atmosphere (e.g., the middle latitude summer, …. The tropical atmosphere). What are the rationalities for this treatment?

5. Line 409-410: "clouds were transformed… " If clouds were predicted by the CMA-MESO (only 19 layers), how to present clouds for the remaining 14 layers? All were set to zero for the last 14 layers?

6. Line 521: "Figure 8. Spectral …..". Does the figure 8 represent a specific atmospheric layer? for the 2m height ? or does it integrate the entire atmospheric profile?

---

## Referee Comment (RC2)

**Review of "Evaluation of radiation schemes in the CMA-MESO model using high time-resolution radiation measurements in China: I. Long-wave radiation" by Yang et al. submitted to Geoscientific Model Development**

This study compares the downward longwave irradiance predicted by the China Meteorological Administration mesoscale model (CMA-MESO) with high-resolution long-term measurements from 42 sites in China. The authors conclude that the model generally agrees with the observations. The correlation between the prediction error and multiple factors (including geolocation, hour of day, cloud amount, etc.) is analyzed. Generally speaking, the dependence of model error on clouds is important and interesting to me. However, the results and the findings presented in current form may not be sufficiently insightful and novel enough to serve as direct feedback to model improvements. In other words, we know that clouds are among the largest contributors to the model uncertainty. Emphasizing it again is not valuable for future model development. It is still necessary to figure out what physical processes specifically contribute to model biases. Approximations made in the radiation scheme? Errors in cloud microphysics? I also have other major concerns regarding the title and the methodology of this study, notably the use of the MODTRAN model. Overall, I recommend a major revision or even a rejection of this version of the manuscript.

Please see below for specific major concerns and minor suggestions.

**Major Concerns**

1. Title: The title of the manuscript is "**Evaluation of radiation schemes** in the CMA-MESO model using high time-resolution radiation measurements in China: I. **Long-wave radiation**". The title is indeed very general and large. Upon reading the title, I would expect the authors will comprehensively compare the model outputs to various radiation measurements (e.g., radiation measured at the top of the atmosphere and at the surface, upwards and downwards). Yet, this study narrows down to only compare the model outputs against the downward longwave irradiance, an important variable though but just one variable in the radiation budget. I would say the title is very misleading, and the authors should polish it to make it more specific and accurate.

In addition, this manuscript seems to be the first part of a multi-part article with the subtitle "I. Long-wave radiation". The authors mentioned in the final section that an evaluation of the shortwave radiation scheme may follow, which is okay. However, for the first part of this article, I would expect an overarching introduction to this model, like what are the highlights of the CMA-MESO model and what technically makes it different from other mesoscale models. What I read in the second paragraph in the introduction is generally the model adopts the RRTM scheme and what the RRTM scheme is. I don't have a picture of what improvements have been made based on this existing radiation scheme, which motivates an evaluation of the model. The authors may also need to include a sentence in the introduction section to inform the readers what should be expected in the next part of the article series.

2. Essentially, this study compares the predictions from the CMA-MESO model to the field measurements. While I understand the difficulty in comparing the gridded model outputs to the sparse and uneven radiation measurements, the authors also introduced a few other changing factors that make the analysis more complicated. Based on my understanding, the authors used the concatenated time series of model data from mixed versions of the model. I am not very sure whether the change is big or small, physical or cosmetic, in each version of the model. However, I do think that such comparison using different versions of model outputs can be challenging and unfair. Different versions of models are evaluated against measurements in different time periods, possibly with variable meteorological conditions. If I were the authors, I would do a "retrospective prediction" using a single version of the model. Say, I could predict a certain time frame in the past records based on the input data before that point. Given that, different versions of the model can be compared to each other, and it would be interesting to see whether one version is better than the others.

Without this complexity, to make it an evaluation of the model, it is important but still difficult to attribute the model error by simply analyzing the statistical correlation between the model bias and the physical variables. For example, the authors claim that the RRTM scheme in the CMA-MESO model struggles with cold and dry conditions. There are so many confounding factors that may lead to a different conclusion. What if those cold and dry samples all come from the plateau region where the model indeed struggles with the topography? An interesting question to answer could be what should be blamed for the difference between the model and observations. The structural error in the radiation scheme? Or the prediction error in clouds/aerosols/temperature/humidity? At the end of this manuscript, the authors mentioned the possibility of checking the model prediction of those meteorological variables with the measurements, which I think should be done in this manuscript to make the argument stronger.

3. Section 3.2.2 tries to elucidate the underlying physical mechanisms that control the downward longwave irradiance using the spectral fluxes from the MODTRAN radiation model. Personally, I think it is basically a radiation 101 material that teaches the readers what factors can affect downward longwave irradiance. It is not relevant to the model evaluation and does not directly explain the model bias. It would be great if the measurements contained spectral fluxes, and then we would be able to diagnose what could be wrong with the model predictions and link it to physical mechanisms. Even the model does not output spectral fluxes that could be used to compare against the MODTRAN simulations. As far as I am concerned, this subsection does not contribute to the main theme of this manuscript and should be entirely removed.

**Minor Suggestions**

L22: "but" I don't think the sentences before and after present contrasting ideas and require a turning point. It might be appropriate to make them separate sentences.

L39-40: "[…] due to the radiation emitted by the instrument body is comparable to that being measured in wavelength […]". First grammatical error "due to" => "because". Second I don't quite understand what this sentence actually means. Please consider rephrasing it.

L42: "Brunt et al., 1932" => "Brunt, 1932". This is a single-author paper.

L57-58: If talking about the longwave fluxes and cooling rates in general, the authors should also consider ground emission. I suggest a simpler version of this sentence: "[…], which are governed by the absorption and emission of the infrared radiation from both the atmosphere and the ground surface (Shen et al., 2004)."

L67: From the perspective of spectral resolution, LBLRTM can be very accurate. But it doesn't sound right to call it the most accurate radiative transfer model. I would say it could be regarded as the baseline model to be compared to.

L77: What is the temporal resolution for those in-situ DnLWI measurements?

Tables 1 & 2: It could be informative to include a map to show the locations of the sites and the instruments each site uses.

L144-145: "There are 2501×1671 grid points in the **north-south** and **east-west** directions, respectively." I suspect there should be 2501 grid points in the east-west direction ($\frac{(145-70)°}{0.03°} + 1$) and 1671 in the north-south direction. Please correct the order.

L149-150: Grammatical error. I don't know which is the main sentence and which is the clause. Please consider rephrasing it.

L169-171: Why should we average ten 1-min DnLWI measurements centered at the punctual hour of the prediction and compare this quantity to hourly instantaneous prediction? What's the temporal step for each integration of the model?

L188: "Talor, 2001" Typo in the author's name.

L195: What does "PDLR" actually stand for? In lines 154-155, the authors explain it as the Prediction Deviation. But where is "LR"? I don't think this is a standard abbreviation and discourage the use of such a confusing term.

L256-257: "The RRTM scheme was remarkably stable when the atmosphere was wet." As I stated in my major concern, there are many confounding factors, and it is hard to argue that humidity directly affects model performance without more sounding proof. Also note the difference in sample size between the bins (18,967 in dry, 330,576 in semi-humid, and 51,738 in humid).

L289-290: "The DnLWIs observed were generally lower than those predicted […] under the cloudless and partly cloudy conditions, but higher than those under the overcast conditions." This is inconsistent with what Fig. 2b-d tells me. Observations (black) are generally higher than predictions (red) except under the overcast conditions.

L297: "[…] mostly come from the insufficient consideration of the PBL in the CMA-MESO model". This argument does not have sufficient supporting evidence, especially considering that much evidence points towards the high and medium clouds, which are located beyond the boundary layer.

L300: "[…] insufficient prediction ability of the RRTM scheme under extremely cold and dry conditions". Again, cold and dry may not be the culprit of the worse model performance.

Figure 2: The y-axis scale is different across different panels. At least for each category (e.g., humidity, season, cloud cover, etc.), the scale should be unified.

L337-339: I don't quite observe the reduced amplitude of PLDR with increasing temperature and surface pressure in Figure 3.

L332: "**All** correlation coefficients". Do they include the negative ones discussed in the previous paragraph?

L355: One caveat here is that the visibility is defined in the shortwave, visible spectrum.

L361: "small correlation coefficient (0.094) between the DnLWI and Vis". The correlation is between PDLR (i.e., the model error in DnLWI) and Vis, not the absolute value of DnLWI and Vis. Also, the sample size and confounding factors matter here.

L384-396: It is valuable to compare predicted meteorological variables to the reanalysis, which can help explain the contribution of model biases. Such should be applied to other variables as well.

L389-391: "CCDs less than –10%". Is it worse than or better than an underestimation of 10%? It could be better phrased as "CCDs are underestimated by greater/smaller than 10%".

L395: "was may relate to" grammatical error.

L403-405: Why T/Q profiles at those vertical layers are filled with values from the standard atmospheres? Are these layers missing in the CMA-MESO model? Then how is radiation treated beyond the 100 hPa level in the CMA-MESO model? It could induce inconsistencies between the CMA-MESO model and the MODTRAN simulations.

L407: "MDDTRAN" typo.

L409-411: Does it mean that only the vertically integrated cloud amount is input to the MODTRAN simulations for the three standard types of clouds? If so, the standard cloud profile defined in MODTRAN for low, medium, and high cloud is scaled and used, which can be very different from the real cloud profiles in the CMA-MESO model. Is my understanding correct?

L412: "anisotropy" This word means the radiation may be stronger in one direction than in others. Do you mean spectral dependency rather than directional dependency?

L439-441: "[…] the reliability of the DnLWIs predicted by the CMA-MESO model may be affected by the inappropriate input of the cloud types, especially the high and medium clouds." Given the input cloud profile described in previous paragraphs, do you mean that the standard cloud profiles defined in the MODTRAN profile are better than the simulated profiles in the CMA-MESO model?

L467-468: "The reason why **uses** […] is **to facilitate detect** the […]." Grammatical errors.

L471: "[…] discrepancies between the RRTM scheme and the radiative transfer model." In my previous comments, I mentioned that the input profiles of both models are not exactly the same. Thus, the comparison of the outputs may not purely present structural differences between the two models.

L490: Delete "e.g."

L491: "[…] they **[are]** closed to zero when the atmosphere altitude **[is]** greater than 10 km." Grammatical errors.

L491: "Not" => "Note"

L493: "[…] the $CO_2$ profiles decreased exponentially with increasing height." It might be worth mentioning the unit of $CO_2$ concentration in the figure.

L574: "[…] due to the strong thermal emission of the clouds". Strong thermal emission of the clouds only explains larger DnLWI under the overcast conditions. It is the model overestimating the longwave cloud radiative effect (LWCRE) that explains the change of model bias from negative to positive.

L589: "uncertain" => "uncertainty".

L590: "are looking forward to" => "are expected to".

---

## Author Comment (AC1)

**Response to Referee 1 Comments**

**General comments:**

1. This paper provides a comprehensive evaluation of the CMA-MESO model's ability to predict downward long-wave irradiance (DnLWI) in China, using extensive high time-resolution in-situ measurements. The study is significant as it identifies the model's performance under various atmospheric conditions and highlights discrepancies particularly under overcast, dry, and cloudless scenarios. The authors also pinpoint the influence of cloud cover and stable nocturnal boundary layer on the model's accuracy. This thorough assessment offers valuable insights for improving the radiation schemes in NWP models and underscores the importance of accurate cloud representation in enhancing model reliability. The manuscript is suitable for publication, once minor revisions are made.

Thank you for this comment, and we are grateful for reviewer's encouraging comments.

2. The title of the manuscript, "Evaluation of radiation schemes in the CMA-MESO model using high time-resolution radiation measurements in China: I. Long-wave radiation", is somewhat misleading. The study exclusively assesses downward long-wave irradiance (DnLWI), without addressing other aspects such as downward long-wave irradiance in specific atmospheric layer or upward long-wave irradiance. The authors can find their suitable words to clarifying this in the title would better reflect the manuscript's focus and scope. Probably, for instance, change "Longwave radiation" to "surface downward Longwave radiation".

Thank you very much for your valuable comments and suggestions.

Exactly, the title of the original manuscript is not very precise, and we would like to modify it to "Evaluation of surface downward long-wave irradiance forecasts from the CMA-MESO V4.1-V5.1 based on high time-resolution radiation measurements in China" in the revision of the manuscript.

**Specific comments:**

1. Line 132: "quality control". The pyrgeometer measurements are only valid within certain spectral ranges. Did this 'quality control' step consider the response function of the pyrgeometer? To validate the LW radiation scheme using in-situ pyrgeometer measurements, signals of DnLWI predictions from the CMA-MESO that are beyond the pyrgeometer's spectral range must be filter out. Did this 'quality control' step consider masking those signals?

Thank you for your mention of effects of the pyrgeometer.

Yes, we have considered the response function of the pyrgeometer before the 'quality control' step. As we all known the initial record of the pyrgeometer is voltage of the sensor, which is subsequently converted into the irradiance via multiplying the sensitivity coefficient of the pyrgeometer. The sensitivity coefficient, which is usually derived from calibration with the reference pyrgeometer, includes the effects of the response function of the sensor, the transmittance of the dome, etc.

Though the radiometers used in this study have spectral ranges of ~4.0 to 50.0 μm and the CMA-MESO model forecasts the DnLWI over a wider spectral range (10-3000 cm$^{-1}$ or 3.33-1000 μm), most of the downward long-wave spectral irradiance emitted by the atmosphere presents within the spectral range of the pyrgeometer according to the Planck's law and the actual atmospheric temperature. Thereby, the portion of DnLWI predicted by the CMA-MESO outside of the spectral range (~4.0 to 50.0 μm) is generally negligible. Even so, we considered this effect to some extent via comparing the DnLWIs observed by the pyrgeometer with the theoretical DnLWI values, which are calculated in terms of the air temperature at 2 m height by using the Stephen-Boltzmann's law during the 'quality control' processing in this study.

2. Lines 161-162: "…… were taken as input parameters into the MODTRAN model", I wonder if the atmospheric profile inputs used to drive the RRTM in the CMA-MESO and MODTRAN are identical? So that comparisons between RRTM and MODTRAN are reasonable to be make.

Thank you for your vital comment.

It is true that the reasonability of comparison between DnLWIs simulated by the MODTRAN model and RRTM in the CMA-MESO model depends on the identical atmospheric profiles. It is worth noting that a height-based-terrain following coordinate proposed by Chen and Somerville (1975) is adopted in the CMA-MESO model (Chen et al., 2008). Specifically speaking, a total of 50 layers of air temperature and relative humidity (or specific humidity) at variable pressure level is used in the CMA-MESO model.

Although the 50-layer air temperature and relative humidity profiles are adopted by the RRTM in the CMA-MESO model, the air temperature and humidity profiles at 19 isobaric surfaces (1000, 975, 950, 925, 900, 850, 800, 750, 700, 650, 600, 550, 500, 450, 400, 350, 300, 200, and 100 hPa) were merely output by the CMA-MESO before 2022. We, thereby, used the output atmospheric profiles at 19 isobaric surfaces together with the background atmospheric profiles (with pressure lower than 100 hPa) to reconstruct the input atmospheric profiles, which were used to drive the MODTRAN model in this study.

To elucidate discrepancies between the original atmospheric profiles used in the CMA-MESO model and the derived atmospheric profiles adopted in the MODTRAN model, we selected five original/derived air temperature and humidity profiles at the XL site under various conditions in 2022 and then put them into the MODTRAN to simulate the DnLWIs. Basic descriptions of the selected cases at the XL site under clear sky, partly

cloudy, and overcast conditions are listed in Table 1, and the air temperature and relative humidity profiles are also plotted in Figure 1.

[Figure]

**Figure 1**. Original/derived atmospheric profiles of the (a) relative humidity and (b) air temperature input into the MODTRAN model for five typical cases at the XL site in 2022. Solid lines and "o" represent the original atmospheric profiles adopted in the CMA-MESO model and the MODTRAN model, and dashed line and "d" represent the derived profiles at the isobaric surfaces used in the MODTRAN model, which stemmed from the output profiles of the CMA-MESO model as well as the background atmospheric profiles.

It can be seen from Figure 1a that the derived relative humidity profiles are generally consistent with the original humidity profiles except that the formers are smoother than the latter at several layers (e.g., 700-500 hPa, etc.). The original air temperature profiles are identical to the derived ones except some negligible discrepancies occurred at the higher levels with the air pressure less than 100 hPa (Fig. 1b). Moreover, the relative deviation between the DnLWI simulated by the MODTRAN model using the original atmospheric profiles and the derived profiles ranges from 0.29% to 0.81% for five cases (Table 1), which gives us a lot of confidence that the comparison between RRTM and MODTRAN in this study is highly reasonable.

**Table1.** Basic descriptions of selected cases at XL site (54102) used to calculate the DnLWIs with the MODTRAN model.

| No. | Time (UTC) | Sky condition | Cloud type | Cloud amount (%) | $T_a$ (K) | Pa (hPa) | PWV (g cm$^{-2}$) | Vis (km) | DnLWI_o (Wm$^{-2}$) | DnLWI_d (Wm$^{-2}$) | Relative deviation (%) |
|---|---|---|---|---|---|---|---|---|---|---|---|
| 1215 | 2022-12-15T05:00 | Clear sky | – | 0 | 256.0 | 892.1 | 0.08 | 24.0 | 135.9 | 136.3 | 0.29 |
| 1007 | 2022-10-07T02:00 | Partly cloudy | High cloud | 55 | 279.6 | 888.7 | 0.54 | 24.0 | 222.7 | 224.5 | 0.81 |
| 0708 | 2022-07-08T10:00 | Partly cloudy | Medium cloud | 49 | 298.1 | 876.3 | 2.14 | 27.8 | 395.1 | 396.6 | 0.38 |
| 0204 | 2022-02-04T02:00 | Partly cloudy | Low cloud | 47 | 254.1 | 893.9 | 0.13 | 24.0 | 221.1 | 221.9 | 0.36 |
| 0801 | 2022-08-01T05:00 | Overcast | High cloud | 100 | 306.4 | 879.7 | 1.51 | 24.0 | 347.5 | 349.9 | 0.69 |

3. L324-331: Figures 2u-y show performances of various versions of the CMA-MESO model. Are there any possible reasons explaining the different performance across various versions of the CMA-MESO model? I recommend the authors to added 2-3 sentences (or probably more sentences) to simply elucidate this somewhere in the text.

Thank you for your valuable comments.

In revision of the manuscript, we added some relevant sentences as well as one reference citation to further elucidate the different performance across various versions of the CMA-MESO model.

"The RRTM scheme in the CMA-MESO model is almost unchanged among different versions, while other improvements in the model may affect its forecasting performance. For instance, improvements in boundary scheme, cloud microphysics scheme, and three-dimensional variational data assimilation system were involved during the evolution of the CMA-MESO version. On the other hand, the humidity field in the CMA-MESO V5.0 is still unstable though it has stable forecast performance with relatively small biases (Ma et al., 2022). The original semi-Lagrangian scheme to calculate the advection of water vapor in the CMA-MESO V5.1 was replaced by an improved material advection scheme, which leads to smaller dispersion and dissipation error in this version of the CMA-MESO model (Peng et al., 2022)".

4. L400-406: "……were replaced with ……the middle latitude summer, the middle latitude winter……" The 88 profiles predicted by the CMA-MESO are 3-h forecast field across 8 hours cycle per day. Right? The total inputs for MODTRAN include 33 layers, this first 19 layers are from CMA-MESO, the remaining 14 layers are from standard atmosphere (e.g., the middle latitude summer, … The tropical atmosphere). What are the rationalities for this treatment?

Thank you for your question.

In this study, the extraction method for all variables predicted by the CMA-MESO (e.g., the DnLWIs, the atmospheric profiles, air temperature at 2-m height, the surface pressure, etc.) is the same. In other word, the CMA-MESO starts forecasting at 12:00 UTC every day, and the later 24-h (12-35 h) hourly punctual forecasts (i.e., at 00-23 UTC on the next day) were selected to be further analyzed. Thereby, the 88 profiles are the randomly selected hourly instantaneous profiles from the later 24-h results predicted by the CMA-MESP that start forecasting at 12:00 UTC.

Previous studies indicate that as much as 60% (90%) of atmospheric emission is derived from the atmosphere within the first 100 m (1 km) under clear-sky conditions. When the sky is overcast, more than 90% originates from within the first 1 km between the ground and the bottom of the cloud (Ohmura, 2001). On the other hand, the CMA-MESO model using 50-layer atmospheric profiles in height-based-terrain following coordinates as an input profile, but it can only output the corresponding atmospheric profiles at 19 isobaric surfaces before 2022. In view of the DnLWI at surface is mainly influenced by the atmosphere within boundary layer, the atmospheric profiles at 19 isobaric surfaces output by the CMA-MESO and the background atmospheric profiles with pressure lower than 100 hPa were used to reconstruct the input atmospheric profiles to drive the MODTRAN model. In addition, the

geographical position of the site and season were also considered during choosing the appropriate background atmospheric profiles (e.g., middle latitude summer, middle latitude winter, etc.).

In short, this treatment is highly trusted to be reasonable according to the test as described in the response to question 2.

  5. Line 409-410: "clouds were transformed…" If clouds were predicted by the CMA-MESO (only 19 layers), how to present clouds for the remaining 14 layers? All were set to zero for the last 14 layers?

Thank you for your comment.

 As mentioned previously (Line 401-402), the minimum pressure of atmospheric profiles in the first 19 layers is 100 hPa, which is approximately15-16 km apart from the ground and reaches the tropopause. As we all know almost all clouds occur in the troposphere, thus, it isn't necessary to present clouds for the remaining 14 layers (i.e., stratosphere and above layers) where the atmospheric water vapor content is very little and cloud is rarely to form.

To improve the accuracy of description, sentence in Line 416-418 is modified as "The cloud options input into the MODTRAN were set in terms of the cloud amount of the high-, medium-, and low-clouds predicted by the CMA-MESO model after matching the corresponding cloud types."

6. Line 521: "Figure 8. Spectral …..". Does the figure 8 represent a specific atmospheric layer? For the 2m height ? or does it integrate the entire atmospheric profile?

Thank you very much for your reminder.

Figure 8 represent the spectral absorption curves of the whole atmosphere, and the title of Figure 8 is modified as "Figure 8. Spectral absorption curves of the *whole* atmosphere that simulated with the MODTRAN model along with atmospheric profiles at the XL site for (a) case 1, (b) case 2, (c) case 3, (d) case 4, and (e) case 5. All absorbing gases and their spectral absorption bands are also denoted".

References

Chen ,D., Xue, J., Yang, X., Zhang, H., Shen, X., Hu, J., Wang, Y., Ji, L., Chen, J.: New generation of multi-scale NWP system (GRAPES): general scientific design. Chin. Sci. Bull., 53, 3433-3445, https://doi.org/10.1007/s11434-008-0494-z, 2008.

Chen ,T. G. and Somerville, R. C. J.: On the use of a coordinate transformation for the solution of the Navier-Stokes equations. J. Comput. Phys., 17, 209-228, https://doi.org/10.1016/0021-9991(75)90037-6, 1975.

Ohmura, A.: Physical basis for the temperature-based melt-index method, J. Appl. Meteorol., 40, 753–761,https://doi.org/10.1175/1520-0450(2001)040%3C0753:PBFTTB%3E2.0.CO;2, 2001.

---

## Author Comment (AC2)

**Response to Chief Editor Comments**

Dear executive editor,

Thank you very much for your reminder and suggestions.

We have read your comments carefully and tried our best to fix the problems as mentioned in the letter. The relevant modifications are addressed as follows.

1. We have updated the dataset (version v2) used in this study and restored it in Zenodo repository at https://doi.org/10.5281/zenodo.12899829. The dataset includes raw data used to evaluate the performance of the CMA-MESO on forecasting the DnLWI as well as several templates (*.sav files), via which the software reads the raw data.

2. We have published our code (IDL programs) as a free software with a GPLv3 license at https://doi.org/10.5281/zenodo.12920314.

3. We have leant the "Code and Data Policy" and core principles (especially item 2) carefully. Due to the copyright license of the CMA-MESO model is managed by the operating management department of the CEMC (Ma et al., 2021), We regrettably have no right to provide the license directly but we provided the contact information of the administrator, i.e., if someone wants to use the CMA-MESO model, he/she can contact the operational management department of the CEMC via email (songzx@cma.gov.cn) or phone (+86-10-68400477).

Therefore, the "*Code and data availability*" part in manuscript is revised as: "The dataset (version v2) used in this study is available online at https://doi.org/10.5281/zenodo.12899829 (Yang and Quan, 2024). Software for evaluating the downward long-wave irradiance predicted by the CMA-MESO model in this study is available at https://doi.org/10.5281/zenodo.12920314, which is the free software with the GPLv3 license (Quan and Yang, 2024). The MODTRAN software is provided as a download at http://modtran.spectral.com/modtran_order. The CMA-MESO model code cannot be distributed due to the copyright license requirement from the CMA Earth System Modelling and Prediction Centre (CEMC). If someone wants to use the CMA-MESO model, he/she can contact the operational management department of the CEMC via email (songzx@cma.gov.cn) or phone (+86-10-68400477)."

4. Exactly, the title of the original manuscript is not very precise, and we would like to modify it to "Evaluation of surface downward long-wave irradiance forecasts from the CMA-MESO V4.1-V5.1 based on high time-resolution radiation measurements in China" in the revision of the manuscript.

References

Ma, Z. S., Zhao, C, F., Gong, J. D., Zhang, J., Li, Z., Sun, J., Liu, Y. Z., Chen, J., and Jiang, Q. G.: Spin-up characteristics with three types of initial fields and the restart effect on

forecast accuracy in the GRAPES global forecast system. Geosci. Model. Dev., 14, 205-221, https://doi.org/10.5194/gmd-14-205-2021, 2021.

Best regards,

Weijun Quan, Martin Wild, and co-authors

---

## Author Comment (AC3)

**Response to Executive Editor Comments**

Dear executive editor,

Thank you very much for your reply.

We are sorry for being two weeks late to response your regarding issues. During these days, we have contacted the administrator of the CEMC several times as well as updated the data set (version 2) and evaluation programs to the repository, which are expected to further improve the manuscript.

1. After all attempts, we received a response from the CEMC as: "*Due to the CMA-MESO is an operational weather forecasting model of the CMA, its license is currently only authorized to the CEMC administrator, anyone who wants to use it can contact the license administrator (songzx@cma.gov.cn; Phone number: +86-10-68400477). In addition, an official website of the CEMC (https://cemc.cma.cn/) is under construction, introduction of the CEMC models in Chinese is now available on it, and the code of the CMA-MESO is expected to be provided in this website before long.*" Currently, we can only get the statement and website address rather than the proof, which is beyond the ability of the administrator.

2. The initial intention of this manuscript is to evaluate the downward longwave irradiance products using the long-term and high precision observation data over various conditions rather than to evaluate the radiation scheme. We regret to write an inappropriate title in the original manuscript, which is inconsistent with the content of the manuscript and leads to misunderstanding. Thus, in the revision of the manuscript, we modify it to "Evaluation of surface downward long-wave irradiance forecasts from the CMA-MESO V4.1-V5.1 based on high time-resolution radiation measurements in China". In this study, we merely used the operational outputs of the CMA-MESO model and related in-situ DnLWI measurements to carry out the evaluation while did not run the CMA-MESO model or modify its code. Thereby, the code of CMA-MESO was not involved in the manuscript.

3. In this study, we used the IDL V8.2 interpreter to compile our programs on evaluating the prediction of the DnLWI from the CMA-MESO. In addition, the .sav files are just simple binary files, which can also be saved in other data format and updated.

Best regards,

Weijun Quan, Martin Wild, and co-authors

2024-08-01

---

## Author Comment (AC4)

**Response to Executive Editor Comments**

Dear executive editor,

Thank you for your reply.

1. We have deposited all data used in this study in zenodo repository (https://zenodo.org/records/13208489), which include the original data, data read template (both in .sav and .txt format), as well as a readme.pdf file to explain all variables in the dataset. The DnLWI data in the outputs from the CMA-MESO model are restored in column "DnLWI_MESO" of the "MESO_RADI_CHN_Hour_DnLWI.csv". Moreover, the software for evaluating the DnLWI predicted by the CMA-MESO model in this study is available at https://doi.org/10.5281/zenodo.12920314, which is the free software with the GPLv3 license. All these data and software are believed to support all the work in this study.

2. We regret to say that we do not have the CMA-MESO code. As mentioned previously, the CMA-MESO model is an operational weather forecast model of CMA and many people are involved in the research and operation of the CMA-MESO. The staffs in CEMC are assigned to different sections (or groups), e.g., model evaluation group, model technology group, data assimilation group, satellite data assimilation group, model coupling group, operational service group, etc.(https://cemc.cma.cn/). Authors of the CEMC in this study belong to the evaluation group of the CEMC, which is responsible for evaluating the products of the CMA-MESO and other CMA models based on in-situ observations or other data, and providing necessary information for the model development group to modify the code. In order to standardize management of the CMA-MESO model, the license is currently only authorized to the CEMC administrator on behalf of the CEMC, anyone who wants to use it can contact the license administrator. According to the law of the CEMC, it is not allowed to distribute the CMA-MESO by others or offer any forms of evidence or proof. The similar situations about license were also declared by other authors in the previous publications in the Geosci. Model Dev.

At last, we would appreciate all reviewers and editors for their valuable comments and suggestions.

Best regards,

Weijun Quan, Martin Wild, and co-authors

2024-08-05

---

## Author Comment (AC6)

**Response to Referee 2 Comments**

**General comments:**

This study compares the downward longwave irradiance predicted by the China Meteorological Administration mesoscale model (CMA-MESO) with high-resolution long-term measurements from 42 sites in China. The authors concluded that the model generally agrees with the observations. The correlation between the prediction error and multiple factors (including geolocation, hour of day, cloud amount, etc.) is analyzed. Generally speaking, the dependence of model error on clouds is important and interesting to me. However, the results and findings presented in current form may not be sufficiently insightful and novel enough to serve as direct feedback to model improvements. In other words, we know that clouds are among the largest contributors to the model improvements. Emphasizing it again is not valuable for future model development. It is still necessary to figure out what physical processes specifically contribute to model biases. Approximations made in the radiation scheme? Error in cloud microphysics? I also have other major concerns regarding the title and the methodology of this study, notably the use of the MODTRAN model. Overall, I recommend a major revision or even a rejection of this version of the manuscript.

Thank you very much for this comment.

The novelty of this study may lie in the fact that the high-temporal (1-min) DnLWI measurements from the pyrgeometer at 42 sites in China are used to evaluate the DnLWI predictions of the CMA-MESO model. As you know, the high-temporal and high quality DnLWI measurements are usually scarce due to the expensive pyrgeometer, and this study is the first time to systematically evaluate the long-term DnLWIs products from the operational CMA-MESO.

Despite the RRTM scheme in the CMA-MESO model has not be evaluated directly in this study, a massive and objective evaluation of the DnLWI products of the CMA-MESO under various conditions is still think to make a sense because the uncertainty of the DnLWI prediction is related to the RRTM scheme as well as the physical processes and the accuracy of the atmospheric profiles used in the model. This work can provide prerequisite information for the Model Technology Section in the CMA Earth System Modelling and Prediction Centre (CEMC) to further check and improve the RRTM scheme in the CMA-MESO model.

Exactly, as you pointed out clouds play important roles in influencing the uncertainty of the DnLWI predicted by the numerical models. From a physical point of view, the emissivity of the atmosphere under all-sky conditions can be derived from that under clear-sky conditions, which is related to the air temperature and vapour pressure, together with a modulation of the cloud. In previous investigations (e.g., Maykut and Church, 1973; Crawford and Duchon, 1999; Duarte et al., 2006; Choi et al., 2008) , the cloud fraction is considered as a vital factor to influence the DnLWI under all-sky

conditions because it can modulate the emissivity of the atmosphere. In this study, we make an effort to show besides the cloud fraction, cloud physical properties (e.g., the cloud types, integrated cloud ice) and other meteorological parameters (e.g., PWV, PBLH) may also lead to the uncertainties of the DnLWI predictions from the CMA-MESO model. To this end, the PLS method is applied in the section 3.2.1, and some conclusions (e.g. the HCC and MCC can bring larger uncertainty of the DnLWI predictions) are expected to offer the useful information to develop the code of the CMA-MESO model.

Please see below for specific major concerns and minor suggestions.

**Major Concerns:**

1. Title: The title of the manuscript is "Evaluation of radiation schemes in the CMA-MESO model using high time-resolution radiation measurements in China:I. Long-wave radiation". The title is indeed very general and large. Upon reading the title, I would expect the authors will comprehensively compare the model outputs to various radiation measurements (e.g., radiation measured at the top of the atmosphere and at the surface, upwards and downwards). Yet this study narrows down to only compare the model outputs against the downward longwave irradiance, an important variable though but just on variable in the radiation budget. I would say the title is very misleading, and the authors should polish it to make it more specific and accurate.

Thank you very much for your valuable comments and suggestions.

As you just pointed out, we regret to put forward an inaccurate title in the original manuscript, which leads to a lot of misunderstanding. We would like to modify it to "Evaluation of surface downward long-wave irradiance forecasts from the CMA-MESO V4.1-V5.1 based on high time-resolution radiation measurements in China" in the revision of the manuscript.

In addition, this manuscript seems to be the first part of a multi-part article with the subtitle "I. Long-wave radiation". The authors mentioned in the final section that an evaluation of the shortwave radiation scheme may follow, which is okay. However, for the first part of this article, I would expect an overarching introduction to this model, like what are the highlights of the CMA-MESO model and what technically makes it different from other mesoscale models. What I read in the second paragraph in the introduction is generally the model adopts the RRTM scheme and what the RRTM scheme is. I don't have a picture of what improvements have been made based on this existing radiation scheme, which motivates an evaluation of the model. The authors may also need to include a sentence in the introduction section to inform the readers what should be expected in the next part of the article series.

Thank you for your comments and suggestions.

The original purpose of our research is to provide the necessary information for the code developers on the performance of the CMA-MESO model over the past years. Thereby,

we have spent a lot of time to process the in-situ radiation observations (including the short- and long-wave radiation) and CMA-MESO products. We also expect to carry out the evaluation of the shortwave radiation part if this work can be finished in the near future.

According to the reviewer's suggestion, the second paragraph in the "Introduction" is updated in the revision as: "The accuracy of the DnLWI predicted by the numerical forecasting models depends largely on both the physical schemes and atmospheric profiles input to the models. […] as well as the rapid radiative transfer model (RRTM, Mlawer et al.,1997), which is employed in the ECMWF model (Morcrette, 2002a), the Weather Research and Forecast model (Iacono et al., 2008), the China Meteorological Administration mesoscale model (CMA-MESO), etc. Note that, some efforts have been made to improve the RRTM scheme used in the CMA-MESO model, e.g., introducing the effects of geographical slopes into the radiation scheme (Xu et al., 2008). The RRTM scheme uses a correlated-k method and look-up tables to accurately compute the long-wave fluxes and cooling rates over the long-wave spectral region (10–3000 $cm^{-1}$ or 3.333–1000 μm), which are governed by the absorption and emission of the infrared radiation from both the atmosphere and the ground surface (Shen et al., 2004). To run the RRTM in the CMA-MESO, some parameters (e.g., the air pressure, temperature, water vapour, mixing ratio of cloud water/ice, cloud cover, the concentrations of $O_3$ and $CO_2$, and the surface emissivity) are required to input, but the effects of aerosols, $N_2O$, $CH_4$, CFC11, and CFC12 are not taken into account. Apart from the RRTM long-wave radiation scheme, other physical parameterization schemes are employed in the CMA-MESO model, e.g., the Dudhia short-wave radiation scheme (Dudhia, 1993), the Noah land surface scheme (Chen and Dudhia, 2001), the MRF planetary boundary layer scheme (Hong and Pan, 1996), and the WSM6 cloud microphysics scheme (Hong et al., 2004). In addition, the CMA-MESO contains both the data assimilation and atmospheric model. The atmospheric model has a fully compressible non-hydrostatic dynamical framework, in which the potential temperature, specific humidity, three-dimensional wind field components, and dimensionless air pressure are utilized as the independent forecast variables (Ma et al., 2022). Moreover, the height-based terrain-following coordinate with the Charney-Philips variable staggering is adopted in vertical direction (Chen et al., 2008). Given the history of the model development, the CMA-MESO is recognized as one of the successors of the Global/Regional Assimilation and PrEdictions System (GRAPES) independently developed by the CMA since the 2000s (Chen et al., 2008; Xu et al., 2008; Xue et al., 2008; Huang et al., 2017; 2022; Ma et al., 2022; Shen et al., 2023)".

2. Essentially, this study compares the predictions from the CMA-MESO model to the field measurements. While I understand the difficulty in comparing the gridded model outputs to the sparse and uneven radiation measurements, the authors also introduced a few other changing factors that make the analysis more complicated. Based on my understanding, the authors used the concatenated time series of model data from mixed versions of the model. I am not very sure whether the change is big or small, physical or cosmetic, in each version of the model. However, I do think that such comparison using different versions of model outputs can be challenging and unfair. Different versions of

models are evaluated against measurements in different time periods, possibly with variable meteorological conditions. If I were the authors, I would do a "retrospective prediction" using a single version of the model. Say, I could predict a certain time frame in the part records based on the input data before that point. Given that, different versions of the model can be compared to each other, and it would be interesting to see whether one version is better than the others.

Thank you for your comments and suggestions.

As you know, in this study, we used the in-situ DnLWI measurements to compare the "historical records" of the DnLWI products forecasted by the operational CMA-MESO model. We consider the above suggestion is interesting as it tries to retrospectively predict the DnLWI using a single version of the model in order to alleviate the unfair in comparison between the different versions of the CMA-MESO model. Nevertheless, the purpose of this study is to reflect the accuracy of the DnLWIs actually predicted by the CMA-MESO model. From this point of view, the "version" of the CMA-MESO in this study more represent the period corresponding to the version running rather than the usual understanding of the version itself. In other words, how accurate of the DnLWI prediction during the running of the version X of the CMA-MESO? Of course, many factors except for the version changing (e.g., the atmospheric profiles) can influence the results.

Without this complexity, to make it an evaluation of the model, it is important but still difficult to attribute the model error by simply analyzing the statistical correlation between the model bias and the physical variables. For example, the authors claim that the RRTM scheme in the CMA-MESO model struggles with cold and dry conditions. There are so many confounding factors that may lead to a different conclusion. What if those cold and dry samples all come from the plateau region where the model indeed struggles with the topography? An interesting question to answer could be what should be blamed for the difference between the model and observations. The structural error in the radiation scheme? or the prediction error in clouds/aerosols/temperature/humidity? At the end of this manuscript, the authors mentioned the possibility of checking the model prediction of those meteorological variables with the measurements, which I think should be done in this manuscript to make the argument stronger.

Thank you for your comments and suggestions.

Just as you mentioned above it is important but still difficult to attribute the model error by simply analyzing the statistical correlation between the model bias and the physical variables, while the correction analysis and the PLS analysis base on the large number of samples can provide us some useful information about which factors may produce large influence on the DnLWI prediction bias. This is expected to help code developers to focus on some most concerned variables.

It is really found in this study that the CMA-MESO model could underestimate the DnLWI in comparison with the DnLWI observed by the pyrgeometers under cold and dry conditions. Just as you pointed out the reasons result in this phenomenon are very complicated. On one hand, the pyrgeometer can overestimate the DnLWI under cold and

dry conditions due to the excessive heat compensation of the radiometer. On the other hand, the CMA-MESO underestimates the DnLWI under cold and dry conditions, which may have relation with the RRTM scheme or its input profiles. Maybe, this interesting puzzle could be uncovered with the aid of spectral DnLWI measurements from the AERI (Atmospheric Emitted Radiance Interferometer).

According to your suggestion, we update the part between L464-L562 in the original manuscript. In the revision, the radio-sounding temperature and humidity profiles at the XL site were compared with the counterparts used in the CMA-MESO model. Moreover, by comparing the spectral DnLWI outputs simulated by the MODTRAN model with the radio-sounding temperature and relative humidity profiles and those from the CMA-MESO model, the influence of the uncertainties of the meteorological variables are explained. See response in L471 in this document for details.

3. Section 3.2.2 tries to elucidate the underlying physical mechanisms that control the downward longwave irradiance using the spectral fluxes from the MODTRAN radiation model. Personally, I think it is basically a radiation 101 material that teaches the readers what factors can affect downward longwave irradiance. It is not relevant to the model evaluation and does not directly explain the model bias. It would be great if the measurements contained spectral fluxes, and then we would be able to diagnose what could be wrong with the model predictions and link it to physical mechanisms. Even the model does not output spectral fluxes that could be used to compare against the MODTRAN simulations. As far as I am concerned, this subsection does not contribute to the main theme of this manuscript and should be entirely removed.

Thank you for your valuable suggestion.

To improve the pertinence of this context, section 3.2.2 is updated in the revision. In other words, the radio-sounding temperature and humidity profiles at the XL site were compared with the counterparts used in the CMA-MESO model. Moreover, by comparing the spectral DnLWI outputs simulated by the MODTRAN model with the radio-sounding temperature and relative humidity profiles and those from the CMA-MESO model, the influence of the uncertainties of the meteorological variables is explained.

See response in L471 in this document for details.

**Minor Suggestions:**

L22: "but" I don't think the sentences before and after present contrasting ideas and require a turning point. It might be appropriate to make them separate sentences.

Thank you for your suggestion.

This is fixed in the revision as: "i.e, with a relative mean bias error (rMBE) of –2.0%. On the other hand, the prediction of the DnLWI is higher than the observed under overcast (3.1%) while lower than that under dry (–5.3%) and cloudless (–5.2%) conditions."

L39-40: "[…] due to the radiation emitted by the instrument body is comparable to that being measured in wavelength […]. First grammatical error "due to"=>"because". Second I don't quite understand what this sentence actually means. Please consider rephrasing it.

Thank you for your comment.

This is fixed in the revision as: "The main reason lies in the fact that the pyrgeometer is usually expensive and sensitive because the radiation emitted by the instrument body is comparable to that being measured in the wavelength according to the Stefan-Boltzmann law and the comparable temperature of the instrument and the object. Thereby, higher requirements are put forward for the manufacture of pyrgeometer in order to overcome the influence of the instrument's own radiation."

L42: "Brunt et al., 1932"=>"Brunt, 1932". This is a single-author paper.

Thank you very much for your reminder. This is fixed in the revision.

L57-58: If talking about the longwave fluxes and cooling rates in general, the authors should also consider ground emission. I suggest a simpler version the this sentence: "[…], which are governed by the absorption and emission of the infrared radiation from both the atmosphere and the ground surface (Shen et al., 2004)"

Thank you for your suggestion.

This is fixed in the revision as: "The RRTM scheme uses a correlated-$k$ method and look-up tables to accurately compute the long-wave fluxes and cooling rates over the long-wave spectral region (10–3000 cm$^{-1}$ or 3.333-1000 µm) , which are governed by the absorption and emission of the infrared radiation from both the atmosphere and the ground surface (Shen et al., 2004)."

L67: From the perspective of spectral resolution, LBLRTM can be very accurate. But it doesn't sound right to call it the most accurate radiative transfer model. I would say it could be regarded as the baseline model to be compared to.

Thank you for your comment.

This is fixed in the revision as: "Though the LBLRTM is regarded as the baseline of radiative transfer model to be compared, it is limited by the precision of the input atmospheric profiles and its complexity for practical applications (Li et al., 2017)."

L77: What is the temporal resolution for those in-situ DnLWI measurements?

Thank you for your comment.

The original temporal resolution for those in-situ DnLWI measurements is 1 minute, while ten 1-min DnLWI measurements centered at the punctual hour were averaged to

represent the observed DnLWI corresponding to the instantaneous DnLWI prediction of the CMA-MESO.

Tables 1 & 2: It could be informative to include a map to show the locations of the sites and the instruments each site uses.

Thank you for your comment.

A map is added in section 2.1 in the revision, in which the locations of the sites as well as the types of instruments installed at these sites are displayed.

[Figure]

**Figure 1**. Geographical locations of 42 radiation sites employed in this study. The purple, green, blue, red, and black dots denote the stations, at which the types of pyrgeometer used are the PIR, HTL-2, IR02, FS-T1, and CG4, respectively.

L144-145: "There are 2501 × 1671 grid points in the **north-south** and **east-west** directions, respectively." I suspect there should be 2501 grid points in the east-west direction ($\frac{(145-70)^o}{0.03^o}+1$) and 1671 in the north-south direction. Please correct the order.

Thank you very much for your reminder. We are sorry to make a writing mistake.

This is fixed in the revision as : "There are 2501×1671 grid points in the east-west and north-south directions, respectively. "

L149-150: Grammatical error. I don't know which is the main sentence and which is the clause. Please consider rephrasing it.

Thank you for your comment.

This is fixed in the revision as: "Eight model cycle forecasts, which have a 36-h integration with hourly outputs, are carried out at 00, 03, 06, 09, 12, 15, 18, and 21 UTC

in a day (Ma et al., 2022). To improve the model's forecasting ability, the cycles at 00 and 12 UTC began to run a 72-h integration after 22 July 2021.

L169-171: Why should we average ten 1-min DnLWI measurements centered at the punctual hour of the prediction and compare this quantity to hourly instantaneous prediction? What's the temporal step for each integration of the model?

Thank you for your question.

As you know, some disturbances may exist in the high-temporal-resolution (e.g., 1 min) DnLWI measurements even the 1-min raw DnLWI data underwent strict quality control test. To alleviate the possible influence caused by the unreasonable 1-min punctual DnLWI measurement in matching the punctual hour DnLWI prediction, ten 1-min DnLWI measurements centered at the punctual hour of the prediction were averaged (smoothed) to represent the observation value corresponding to the DnLWI predicted.

The temporal step for each integration of the model is 30 s .

L188: "Talor, 2001" Typo in the author's name.

Thank you for your careful inspection.

This is fixed in the revision as: "Ta**y**lor, 2001"

L195: What does "PDLR" actually stand for? In lines 154-155, the authors explain it as the Prediction Deviation. But Where is "LR"? I don't think this is a standard abbreviation and discourage the use of such a confusing term.

Thank you for your question and suggestion.

For simplicity and understandability, a new abbreviation is defined in lines 154-155 as: "…the prediction deviation of the downward long-wave irradiance (PDDLI) from the CMA-MESO model…". At the same time, all "PDLR" in the manuscript are replaced by the "PDDLI".

L256-257: "The RRTM scheme was remarkably stable when the atmosphere was wet." As I stated in my major concern, there are many confounding factors, and it is hard to argue that humidity directly affects model performance without more sounding proof. Also note the difference in sample size between the bins (18, 967 in dry, 330, 576 in semi-humid, and 51,738 in humid).

Thank you for your comments.

Though difference in sample size exists among bins under various conditions, the statistical results for each condition should be reliable due to the use of a large number of samples in statistical calculation. In this study, we did find a phenomenon that the DnLWI predicted by the CMA-MESO is closer to the measurement under humid

conditions, which may be related to the performance of the RRTM scheme but it is not entirely dependent on the scheme.

Thereby, the sentence in L256-257 is rewritten as: "Meanwhile, a smaller rRMSE (5.1%) under humid conditions implied that the DnLWI predicted by the CMA-MESO model was closer to the observation when the atmosphere is wet, which may be related to the performance of the RRTM scheme as well as the water vapor amount in the atmosphere."

L289-290: "The DnLWI observed were generally lower than those predicted […] under the cloudless and partly cloudy conditions, but higher than those under the overcast conditions." This is inconsistent with what Fig. 2b-d tells me. Observations (black) are generally higher than predictions (red) except under the overcast conditions.

Thank you for your careful reminder.

I'm sorry we made a typing mistake, and this is fixed in the revision as: "The DnLWIs predicted by the CMA-MESO model under the cloudless and partly cloudy conditions were generally lower than those observed (Fig. 2b and 2c), but higher than those under the overcast conditions (Fig. 2d)."

L297: "[…] mostly come from the insufficient consideration of the PBL in the CMA-MESO model". This argument does not have sufficient supporting evidence, especially considering that much evidence points towards the high and medium clouds, which are located beyond the boundary layer.

Thank you for your comment.

The sentence in L296-L297 is fixed in the revision as: "In contrast, an apparent underestimation of the predicted DnLWI under dry conditions during the daytime (rRMBE= –7.6%) may come from the insufficient estimation of water vapor amount in the CMA-MESO model."

L300: "[…] insufficient prediction ability of the RRTM scheme under extremely cold and dry conditions". Again, cold and dry may not be the culprit of the worse model performance.

Thank you for your comment.

It is really found that the CMA-MESO model could underestimate the DnLWI in comparison with the DnLWI observed by the pyrgeometers under cold and dry conditions. Just as you pointed out the reasons result in this phenomenon are very complicated. On one hand, the pyrgeometer can overestimate the DnLWI under cold and dry conditions due to the excessive heat compensation of the radiometer. On the other hand, the underestimation of the DnLWI from the CMA-MESO under cold and dry conditions may relate with the RRTM scheme or its input profiles, but lack of the direct

evidence. Maybe, this interesting puzzle could be uncovered with the aid of spectral DnLWI measurements from the AERI (Atmospheric Emitted Radiance Interferometer).

The sentence in L299-301 is modified in the revision as: "The CMA-MESO model considerably underestimated the DnLWIs (with an rMBE of –3%) in winter (Fig. 2k), whereas, matched the observations very well at night in other seasons (Fig. 2h–2j)."

Figure 2: The y-axis scale is different across different panels. At least for each category (e.g., humidity, season, cloud cover, etc.), the scale should be unified.

Thank you for your suggestion.

We have redrawn the Figure 2 with a unified y-axis scale range of 90 W m$^{-2}$, via which the difference between the observed and predicted DnLWIs under various conditions can be seen directly.

[Figure]

**Figure 2.** Diurnal variation of mean hourly DnLWI observations (denoted with the black lines with dots) and predictions from all versions of the CMA-MESO model (denoted with the red lines with circles).

L337-339: I don't quite observe the reduced amplitude of PLDR with increasing temperature and surface pressure in Figure 3.

Thank you for your comment.

Yes, the variation of the amplitude of PDDLI against the temperature and surface pressure is too small to discern in Fig. 3a and b. The sentence "On the other hand, with increasing temperature and surface pressure, the amplitude of the PDLR slightly decreased, which implies that the CMA-MESO model could yield relatively stable DnLWI over the warm and low altitude areas" in the original manuscript is removed in the revision.

L352: "All correlation coefficients". Do they include the negative ones discussed in the previous paragraph?

Thank you for your reminder.

Yes, they include the negative ones in the previous paragraph. This is fixed in the revision as: "Both the positive and negative correlation coefficients were significant at the 95% confidence level."

L355: One caveat here is that the visibility is defined in the shortwave, visible spectrum.

Thank you very much for your valued reminder.

Sentences between Line 355 and 361 in the manuscript are reorganized in the revision as: "In addition, aerosols can heat the Earth's surface through enhancing the downwelling long-wave radiation, which depends on the types, concentration, and height of the aerosol (Zhou and Savijärvi, 2014). Unfortunately, due to the lack of chemical transmission module in the CMA-MESO model, the effect of aerosols is not abundantly considered during predicting the DnLWI. To roughly illustrate the potential influence of aerosol effects on the PDDLI, a correlation coefficient (0.094) between the PDDLI and Vis, which is considered as an indirect indicator of the aerosol (Elterman, 1970; Kaufman and Fraser, 1983; Wu et al., 2014), was also calculated in this study (Fig. 3f)."

L361: "small correlation coefficient (0.094) between the DnLWI and Vis". The correlation is between PDLR (i.e., the model error in DnLWI) and Vis, not the absolute value of DnLWI and Vis. Also, the sample size and confounding factors matter here.

Thank you for your reminder.

This is fixed in the revision. Please see the previous amendment (L355).

L384-396: It is valuable to compare predicted meteorological variables to the reanalysis, which can help explain the contribution of model biases. Such should be applied to other variables as well.

Thank you for your comment.

As the high-temporal-resolution cloud properties (e.g., cloud amount, cloud types, etc.) are usually scarce at meteorological station, we used the ERA5 atmospheric reanalysis products to evaluate the cloud cover predicted by the CMA-MESO model in this study. As for other variable especially the vita variables to DnLWI (e.g., temperature profile, humidity profile), we could compare some model output and real sounding profiles at the XL to further elucidate the influence of the meteorological variables on DnLWI (see the amendment in the Section 3.2.2 in the revision).

L389-391: "CCDs less than –10%". Is it worse than or better than an underestimation 10% ? It could be better phrased as "CCDs are underestimated by greater/smaller than 10%".

Thank you for your suggestion.

 "CCDs less than –10%" means it is worse than an underestimation 10%.

According to you suggestion, we have revised the relevant statements (L389-396 in the original manuscript) as: "The results showed that the CMA-MESO model underestimated the TCC and MCC (with the CCDs underestimated greater than 10%) over most regions of China. The CCDs of the HCC were underestimated greater than 5% over whole China except for the Qinhai-Tibet Plateau, over which the CCDs were overestimated greater than 5%. However, the LCC predicted by the CMA-MESO model were relatively unbiased, i.e., the CCDs of the LCC varied within ±5% over most regions of China."

L395: "was may relate to" grammatical error.

Thank you for your reminder.

This grammatical error is fixed in the revision.

L403-405: "Why T/Q profiles at those vertical layers are filled with values from the standard atmospheres? Are these layers missing in the CMA-MESO model? Then how is radiation treated beyond the 100 hPa level in the CMA-MESO model? It could induce inconsistencies between the CMA-MESO model and the MODTRAN simulations.

Thank you for your questions.

Firstly, a total of 50-layer T/Q profiles are used by the RRTM in the CMA-MESO model. Note that a height-based-terrain following coordinate is adopted in the CMA-MESO model, and the height of the 50 layers range from elevation of the ground to ~ 34 km (or ~ 7 hPa). These layers never missing in the CMA-MESO model, and the radiation of layers beyond the 100 hPa are also treated in the CMA-MESO model.

Whereas, the air temperature and humidity profiles at 19 isobaric surfaces (1000, 975, 950, 925, 900, 850, 800, 750, 700, 650, 600, 550, 500, 450, 400, 350, 300, 200, and 100

hPa) were merely output by the CMA-MESO before 2022. So, we used the output atmospheric profiles at 19 isobaric surfaces from the CMA-MESO together with the background atmospheric profiles (with pressure lower than 100 hPa) to reconstruct the input atmospheric profiles, which were used to drive the MODTRAN model in this study.

To elucidate discrepancies between the original atmospheric profiles used in the CMA-MESO model and the derived atmospheric profiles adopted in the MODTRAN model, we selected five original/derived profiles of air temperature and humidity at the XL site under various conditions in 2022 and then put them into the MODTRAN to simulate the DnLWIs. Basic descriptions of the selected cases at the XL site under clear sky, partly cloudy, and overcast conditions are listed in Table 1, and the air temperature and relative humidity profiles are also plotted in Figure 3.

[Figure]

**Figure 3**. Original/derived atmospheric profiles of the (a) relative humidity and (b) air temperature input into the MODTRAN model for five typical cases at the XL site in 2022. Solid lines and "o" represent the original atmospheric profiles adopted in the CMA-MESO model and the MODTRAN model, and dashed line and "d" represent the derived profiles at the isobaric surfaces used in the MODTRAN model, which stemmed from the output profiles of the CMA-MESO model as well as the background atmospheric profiles.

It can be seen from Figure 3a that the derived relative humidity profiles are generally consistent with the original humidity profiles except that the formers are smoother than the latter at several layers (e.g., 700-500 hPa, etc.). The original air temperature profiles are identical to the derived ones except some negligible discrepancies occurred at the higher levels with the air pressure less than 100 hPa (Fig. 1b). Moreover, the relative deviation between the DnLWI simulated by the MODTRAN model using the original atmospheric profiles and the derived profiles ranges from 0.29% to 0.81% for five cases (Table 1), which gives us a lot of confidence that the comparison between RRTM and MODTRAN in this study is highly reasonable.

**Table1.** Basic descriptions of selected cases at XL site (54102) used to calculate the DnLWIs with the MODTRAN model.

| No. | Time (UTC) | Sky condition | Cloud type | Cloud amount (%) | $T_a$ (K) | Pa (hPa) | PWV (g cm$^{-2}$) | Vis (km) | DnLWI_o (Wm$^{-2}$) | DnLWI_d (Wm$^{-2}$) | Relative deviation (%) |
|---|---|---|---|---|---|---|---|---|---|---|---|
| 1215 | 2022-12-15T05:00 | Clear sky | – | 0 | 256.0 | 892.1 | 0.08 | 24.0 | 135.9 | 136.3 | 0.29 |
| 1007 | 2022-10-07T02:00 | Partly cloudy | High cloud | 55 | 279.6 | 888.7 | 0.54 | 24.0 | 222.7 | 224.5 | 0.81 |
| 0708 | 2022-07-08T10:00 | Partly cloudy | Medium cloud | 49 | 298.1 | 876.3 | 2.14 | 27.8 | 395.1 | 396.6 | 0.38 |
| 0204 | 2022-02-04T02:00 | Partly cloudy | Low cloud | 47 | 254.1 | 893.9 | 0.13 | 24.0 | 221.1 | 221.9 | 0.36 |
| 0801 | 2022-08-01T05:00 | Overcast | High cloud | 100 | 306.4 | 879.7 | 1.51 | 24.0 | 347.5 | 349.9 | 0.69 |

L407: "MDDTRAN" typo.

We are sorry to make a spell mistake.

This is fixed in the revision as: "MODTRAN"

L409-411: Does it mean that only the vertically integrated cloud amount is input to the MODTRAN simulations for the three standard types of clouds? If so, the standard cloud profile defined in MODTRAN for low, medium, and high cloud is scaled and used, which can be very different from the real cloud profiles in the CMA-MESO model. Is my understanding correct?

Thank you for your question.

Firstly, cloud profiles are used in the CMA-MESO model, while the output of the cloud amounts in whole atmosphere for various cloud types are used in the MODTRAN. In CMA-MESO, a post-processing algorithm is applied to cloud profiles to derive the integrated cloud amounts for various cloud types in terms of the cloud profiles according to the cloud top/base pressures.

**Table2.** Thresholds for various cloud types output from the CMA-MESO model

| Cloud type | Cloud top pressure (hPa) | Cloud bottom pressure (hPa) |
|---|---|---|
| high cloud | 50.0 | 440.0 |
| medium cloud | 440.0 | 680.0 |
| low cloud | 680.0 | 1200.0 |

In MODTRAN, the cloud model for whole atmosphere rather than the cloud profile is required to input. For instance,

**Table3.** Thresholds for various cloud types input to the MODTRAN model

| Cloud type | Cloud top altitude (km) | Cloud bottom altitude (km) |
|---|---|---|
| high cloud | depends on geolocation and season | 6.5 |
| standard cirrus | – | 6.5 |
| medium cloud | 6.5 | 2.0 |
| altostratus cloud | 3.0 | 2.4 |
| low cloud | 2.0 | 0.0 |
| stratus cloud | 1.0 | 0.33 |
| strato-cumulus | 2.0 | 0.66 |
| nimbostratus | 0.66 | 0.16 |

Secondly, the integrated cloud amounts for various cloud type output from the CMA-MESO model were matched to the corresponding input cloud amounts according to the pressure/altitude of the cloud top/bottom, which is used to drive the MODTRAN.

L412: "anisotropy" This word means the radiation may be stronger in one direction than in others. Do you mean spectral dependency rather than directional dependency?

Thank you for your suggestion.

This is fixed in the revision as: "a *tape*8 file consists of spectral irradiance data, in which the directionality of the downwelling long-wave radiation is well taken into account."

In my opinion, it is difficult to state whether the spectral dependency or the directional dependency to bring greater uncertainty during simulating the DnLWI since they belong to two different aspects. However, this work gave us a deep learning that the effect of the direction on the long-wave radiation propagation is very important via comparing the irradiance results from the integration of the radiance with/without directions consideration.

L439-441: "[…] the reliability of the DnLWIs predicted by the CMA-MESO model may be affected by the inappropriate input of the cloud types, especially the high and medium clouds." Given the input cloud profile described in previous paragraphs, do you mean the standard cloud profiles defined in the MODTRAN profile are better than the simulated profiles in the CMA-MESO model?

Thank you for your question.

This issue has been explained in detail in the response to L409-411.

L467-468: "The reason why **uses** […] is **to facilitate detect** the […]." Grammatical errors.

Thank you for your reminder.

This is fixed in the revision.

L471: "[…] discrepancies between the RRTM scheme and the radiative transfer model." In my previous comments, I mentioned that the input profiles of both models are not exactly the same. Thus, the comparison of the outputs may not purely present structural differences between the two models.

Thank you for your comment.

The paragraph in L464-562 in the original manuscript is replaced by the following paragraph in the revision to further explain the potential influence of the temperature and humidity profiles used in the CMA-MESO model on the DnLWI predictions. To this end, the MODTRAN model is applied together with the air temperature/humidity profile data from both the CMA-MESO model and the radiosonde at XL to simulate the DnLWIs.

"To further elucidate the potential influence of the temperature/humidity profiles used in the CMA-MESO on the DnLWI predictions, three typical cases occurred at XL were selected to analyze. Figure 4 shows the profiles of the temperature and relative humidity, which were derived from the CMA-MESO intrinsic profiles as well as those from the radiosonde carried out at the XL station at 00 UTC 23 August 2021 (Fig. 4a, 4d), 00 UTC 31 May 2021 (Fig. 4b,

4e), and 00 UTC 8 May 2021 (Fig. 4c, 4f).In general, the air temperature profiles used in the CMA-MESO model consist well with those from the radiosonde in three cases except in the case occurred at 00 UTC 31 May 2021, in which the temperature from the radiosonde remarkably less than those from the CMA-MESO mode between the ground and the layer at 300 hPa (Figure 4b). On the other hand, the humidity profiles from the CMA-MESO are obviously different from those derived from the radiosonde (Fig. 4d-f), In particular, the relative humidity profile of the CMA-MESO is quite different from the counterpart from the radiosonde (Fig. 4e).

[Figure]

**Figure 4**. Temperature profiles for cases occurred at XL (a) at 20210823T00:00 UTC, (b) 20210531T00:00 UTC, and (c) 20210508T00:00 UTC, respectively. The corresponding relative humidity profiles for these cases are plotted in (d), (e), and (f). The black curves with dots denote the atmospheric profiles derived from the radiosonde, and the red curves with circles denote ones used in the CMA-MESO model.

"The basic descriptions of these cases and the relevant results of the DnLWIs are listed in Table 4. The 'Ta/Pa' represent the air temperature/Pressure at 2-m height above the ground, and the 'PWV' represents the precipitable water vapor amount. In addition, the 'OBS', 'MESO', 'MOD_d', and 'MOD_r' represent the DnLWI observed by the pyrgeometer, predicted by the CMA-MESO model, and those simulated by the MODTRAN model with the atmospheric profiles from both the CMA-MESO and the radiosonde as input variables, respectively. It can be seen from Fig. 4a and Table4, the MODTRAN model can produce more closer DnLWIs (294.3 and 293.0 W m$^{-2}$) to the observations in compare with the CMA-MESO model, but its overestimation/underestimation of the DnLWIs depends on the

'warmer'/ 'cooler' temperature profiles. Note that the very close temperature profiles and relative close relative humidity occurred at 00 UTC 8 May 2021 (Fig. 4c, 4f) results in the similar DnLWIs (309.9 and 308.4 W m$^{-2}$) simulated by the MODTRAN model. The largest difference between the temperature and humidity profiles from the CMA-MESO model and the radiosonde can result in a more than 33 W m$^{-2}$ in DnLWIs simulations using the MODTRAN model (Fig. 4b and 4e). Therefore, the influence of the atmospheric profiles on the DnLWI prediction of the CMA-MESO model should be considered in the future work.

**Table4.** Basic descriptions of selected cases at XL site (54102) used to calculate the DnLWIs with the MODTRAN model along with the atmospheric profiles both derived from the CMA-MESO model and from the radiosonde carried out at XL.

| Time (UTC) | Sky condition | Cloud type | Cloud amount (%) | $T_a$ (K) | Pa (hPa) | PWV (g cm$^{-2}$) | DnLWI (Wm$^{-2}$) | | | |
|---|---|---|---|---|---|---|---|---|---|---|
| | | | | | | | OBS | MESO | MOD_d | MOD_r |
| 2021-08-23T00:00 | Clear sky | – | 0 | 290.2 | 877.9 | 1.52 | 293.6 | 295.4 | 293.0 | 294.3 |
| 2021-05-31T00:00 | Partly cloudy | Medium cloud | 63 | 290.3 | 872.4 | 1.85 | 368.3 | 365.0 | 363.8 | 330.5 |
| 2021-05-08T00:00 | Overcast | Low cloud | 100 | 275.8 | 876.8 | 0.68 | 314.3 | 313.9 | 309.9 | 308.4 |

L490: Delete "e.g."

Thank you for your reminder.

This is fixed in the revision.

L491: "[…] they [are] closed to zero when the atmosphere altitude [is] greater than 10 km." Grammatical errors.

Thank you for your suggestion.

This is fixed in the revision.

L491: "Not"=>"Note".

Thank you for your reminder.

This is fixed in the revision.

L493: "[…] the $CO_2$ profiles decreased exponentially with increasing height" It might be worth mentioning the unit of $CO_2$ concentration in the figure.

Thank you for your reminder.

The unit of CO2 concentration in Fig. 7d is right, which is a conversion from the input $CO_2$ mixture ratio (ppmv) in the MODTRAN.

L574: "[…] due to the strong thermal emission of the clouds". Strong thermal emission of the clouds only explains larger DnLWI under the overcast conditions. It is the model overestimating the longwave cloud radiative effect (LWCRE) that explains the change of model bias from negative to positive.

Thank you very much for your valuable suggestion.

This is fixed in the revision as: "[…] due to the CMA-MESO overestimating the longwave cloud radiative effect under cloudy conditions."

L589: "uncertain"=> "uncertainty".

Thank you for your reminder.

This is fixed in the revision as: "uncertainty"

L590: "are looking forward to"=> "are expected to".

Thank you for your reminder.

This is fixed in the revision.

**Reference**

Chen, F. and Dudhia, J.: Coupling an advanced land surface-hydrology model with the penn state-NCAR MM5 modeling system. Part I: model implementation and sensitivity, Mon. Weather Rev., 129, 569–585, https://doi.org/10.1175/1520-0493(2001)129%3C0569:CAALSH%3E2.0.CO;2, 2001.

Choi, M., Jacobs, J. M., and Kustas, W. P.: Assessment of clear and cloudy sky parameterizations for daily downwelling longwave radiation over different land surfaces in Florida, USA, Geophys. Res. Lett., 35, L20402, https://doi.org/10.1029/2008GL035731, 2008.

Crawford, T. M. and Duchon, C. E.: An improved parameterization for estimating effective atmospheric emissivity for use in calculating daytime downwelling longwave radiation, J. Appl. Meteorol., 38, 474-480, 1999.

Duarte, H. F., Dias, N. L., and Maggiotto, S. R.: Assessing daytime downward longwave radiation estimates for clear and cloudy skies in Southern Brazil, Agr. Forest. Meteorol., 139, 171-181, https://doi.org/10.1016/j.agrformet.2006.06.008, 2006.

Dudhia, J.: A nonhydrostatic version of the Penn State-NCAR mesoscale model: Validation tests and simulation of an Atlantic cyclone and cold front, Mon. Weather Rev., 121, 1493-1513,

https://doi.org/10.1175/1520-0493(1993)121%3C1493:ANVOTP%3E2.0.CO;2, 1993.

Hong, S. Y., Dudhia, J., and Chen, S. H.: A revised approach to ice microphysical processes for the Bulk parameterization of clouds and precipitation, Mon. Weather Rev., 132, 103-120, https://doi.org/10.1175/1520-0493(2004)132%3C0103:ARATIM%3E2.0.CO;2, 2004.

Hong, S. Y. and Pan, H. L.: Nonlocal boundary layer vertical diffusion in medium-range forecast model, Mon. Weather Rev., 124, 2322-2339, https://doi.org/10.1175/1520-0493(1996)124%3C2322:NBLVDI%3E2.0.CO;2, 1996.